# Achieving Budget-optimality with Adaptive Schemes in Crowdsourcing

**Ashish Khetan and Sewoong Oh**
Department of ISE, University of Illinois at Urbana-Champaign
Email: {khetan2,swoh}@illinois.edu

## Abstract

Adaptive schemes, where tasks are assigned based on the data collected thus far, are widely used in practical crowdsourcing systems to efficiently allocate the budget. However, existing theoretical analyses of crowdsourcing systems suggest that the gain of adaptive task assignments is minimal. To bridge this gap, we investigate this question under a strictly more general probabilistic model, which has been recently introduced to model practical crowdsourcing datasets. Under this generalized Dawid-Skene model, we characterize the fundamental trade-off between budget and accuracy. We introduce a novel adaptive scheme that matches this fundamental limit. A given budget is allocated over multiple rounds. In each round, a subset of tasks with high enough confidence are classified, and increasing budget is allocated on remaining ones that are potentially more difficult. On each round, decisions are made based on the leading eigenvector of (weighted) non-backtracking operator corresponding to the bipartite assignment graph. We further quantify the gain of adaptivity, by comparing the tradeoff with the one for non-adaptive schemes, and confirm that the gain is significant and can be made arbitrarily large depending on the distribution of the difficulty level of the tasks at hand.

## 1 Introduction

Crowdsourcing platforms provide labor markets in which pieces of micro-tasks are electronically distributed to a pool of workers. In typical crowdsourcing scenarios, such as those on Amazon's Mechanical Turk, a requester posts a collection of tasks, and a batch is picked up by any worker who is willing to complete it. The worker is subsequently rewarded for each task he/she completes. However, some workers are spammers trying to make easy money. Moreover, since the reward is small and tasks are tedious, errors are common even among those who try. To correct for the errors, a common approach is to introduce redundancy by assigning each task to multiple workers and aggregating their responses using some schemes such as majority voting.

A fundamental problem of interest is how to maximize the accuracy of thus inferred solutions, while using as small number of repetitions as possible. There are two challenges in achieving such an optimal tradeoff between accuracy and the budget: $(a)$ we need a scheme for deciding which tasks to assign to which workers; and $(b)$ at the same time infer the true solutions from their responses.

Since the workers are fleeting, the requester has no control over who gets to work on which tasks. It is impossible to make a trust relationship with the workers. In particular, it does not make sense to explore reliable workers, and exploit them in subsequent steps. Each arriving worker is completely new and you may never get him back. Nevertheless, by comparing responses from multiple workers, we can estimate the true answer to the task, and use it in subsequent steps to learn the reliability of the workers. Our beliefs on the true answers as well as the difficulty of the tasks and the reliability of the workers can be iteratively refined, and one can potentially choose to assign more workers to the more difficult tasks. We would like to understand such intricate interplay of task assignment and inference.

**Setup.** We have $m$ binary classification tasks to be completed by workers. We assume a recent generalization of the Dawid-Skene model introduced in [22] to model the responses, which captures the heterogeneity in the tasks as well as the workers. Precisely, each new arriving worker is parametrized by a quality parameter $p_j \in [0, 1]$ (for the $j$-th arriving worker), which is i.i.d. according to some prior distribution $\mathscr{F}$. Each task is parametrized by a difficulty parameter $q_i \in [0, 1]$ (for the $i$-th task), which is drawn i.i.d. according to some prior distribution $\mathscr{G}$. When a worker $j$ is assigned a task $i$, the task is perceived as a positive task with probability $q_i$, and as a negative task otherwise. Hence, if $q_i$ is close to a half then it is confusing and difficult to correctly classify, and easy if close to one or zero. When task $i$ is assigned to worker $j$, the response is a noisy perception of the task:

$$A_{ij} = \begin{cases} 1, & \text{w.p.} \quad q_i p_j + \bar{q}_i \bar{p}_j, \\ -1, & \text{w.p.} \quad \bar{q}_i p_j + q_i \bar{p}_j. \end{cases} \tag{1}$$

where $\bar{q}_i = 1 - q_i$ and $\bar{p}_j = 1 - p_j$. With probability $p_j$, the worker answers truthfully as he perceives the task, and otherwise gives the opposite answer. Hence, if $p_j$ is close to one then he tells the truth (in his opinion) and if it is close to half he gives random answers. If it is zero, he is also reliable, in the sense that a requester who can correctly decode his reliability can extract the truths exactly.

We define the ground truth of a task as what the majority of the workers agree on, had we asked all the workers. Accordingly, we assume that $\mathbb{E}_{\mathscr{F}}[p_j] > 1/2$ and the true labels are defined as $t_i = \mathbb{I}_{\{q_i > (1/2)\}} - \mathbb{I}_{\{q_i < (1/2)\}}$. Otherwise, we do not impose any condition on the distribution of $p_j$'s. However, we assume $q_i$'s are discrete random variables with support at $K$ points. Our results do not directly depend on this support size $K$, and therefore $K$ can be made arbitrarily large. Note that we focus on only binary tasks with two types of classes, and also the workers are assumed to be symmetric, i.e. the error probability does not depend on the perceived label of the task. The original Dawid-Skene model introduced in [3] and analyzed in [9] is a special case, when all tasks are equally easy, i.e. $q_i$'s are either one or zero. This makes inference easier as all tasks are perceived their true class; the only source of error is in workers' noisy responses.

We assume the following task assignment scenario to model practical crowdsourcing systems. It is a discrete time system, where at the beginning of each time step the requester can create a batch of tasks. This batch is picked up by a new arriving worker, and his/her responses are collected. To model real-world constraints we assume there is a limit on how many tasks a single worker can complete, which we denote by $r$. The requester (also called task master) has no control over who is arriving next, but he has control over which of the $m$ tasks are to be solved by the next arriving worker. This allows for adaptive task assignment schemes, where the requester can choose to include those tasks that he is most uncertain about based on all the history of responses collected thus far.

We consider all randomized task assignment schemes, whose expected number of assignment per task is $\ell$, and all inference algorithms. We study the minimax rate when the nature chooses the worst case priors $\mathscr{F}$ and $\mathscr{G}$ (from a family of priors parametrized by average worker reliability $\beta$ and average task difficulty $\lambda$ defined in (2)), and we choose the best possible adaptive task assignment together with the best possible inference algorithm. We further propose a novel adaptive approach that achieves this minimax rate up to a constant factor. Our approach is different from existing adaptive schemes in [5], where there are multiple types of tasks and the main source of uncertainty is which type the next arriving worker is expert on. Golden tasks with known answers are used to explore expertise and tasks are assigned accordingly.

**Related work.** Existing work on crowdsourcing systems study the standard Dawid-Skene (DS) model [3], where all tasks are equally difficult and hence $q_i \in \{0, 1\}$ for all tasks. Several inference algorithms have been proposed [3, 17, 6, 16, 4, 7, 11, 23, 10, 21, 2, 8, 14], and the question of task assignment is addressed in [9], where the minimax rate on the probability of error is characterized and a matching task assignment scheme and an inference algorithm are proposed. Perhaps surprisingly, for the standard DS model, a non-adaptive task assignment scheme achieves the fundamental limit. Namely, given $m$ tasks and a total budget for $m\ell$ responses, the requester first constructs a bipartite task-assignment graph with $m$ task nodes, $n = m\ell/r$ worker nodes, and edges drawn uniformly at random with degree $\ell$ for the task nodes and $r$ for the worker nodes. Then, $j$-th arriving worker is assigned a batch of $r$ tasks that are adjacent to the $j$-th worker node. Together with an inference algorithm explained in detail in Section 2, this achieves a near-optimal performance. Namely, to achieve an average probability of error $\varepsilon$, it is sufficient to have total budget $O((m/\beta) \log(1/\varepsilon))$, where $\beta = \mathbb{E}_{\mathscr{F}}[(2p_j - 1)^2]$ is the quality of the workers defined in (2). Perhaps surprisingly, no adaptive assignment can improve upon it. Even the best adaptive scheme and the best inference

algorithm still requires $\Omega((m/\beta)\log(1/\varepsilon))$ total budget. Hence, there is no gain in adaptivity. This negative result relies crucially in the fact that under the standard DS model, all tasks are inherently equally difficult. Hence, adaptively assigning more workers to relatively more ambiguous tasks has only a marginal gain. However, simple adaptive schemes are widely used in practice, where significant gains are achieved; in real-world systems, tasks are widely heterogeneous. To capture such varying difficulties in the tasks, generalizations of the DS model were proposed in [19, 18, 22, 15] and significant improvement has been reported on inference problems for real datasets.

The generalized DS model serves as the missing piece in bridging the gap between practical gains of adaptivity and theoretical limitations of adaptivity. We investigate the fundamental question of "do adaptive task assignments improve accuracy?" under this generalized Dawid-Skene model of Eq. (1).

**Contributions.** To investigate the gain of adaptivity, we first characterize the fundamental lower bound on the budget required to achieve a target accuracy. To match this fundamental limit, we introduce a novel *adaptive* task assignment scheme. Our approach consists of multiple rounds of non-adaptive schemes, and we provide sharp analyses on the performance at each round, which guides the design of the task assignment in each round adaptively using the data from previous rounds. The proposed adaptive task assignment is simple to apply in practice, and numerical simulations confirm the superiority compared to state-of-the-art non-adaptive schemes. Under a certain assumption on the choice of parameters in the algorithm, which requires a moderate access to an oracle, we can prove that the performance of the proposed adaptive scheme matches that of the fundamental limit up to a constant factor. Finally, we quantify the gain of adaptivity by proving a strictly larger lower bound on the budget required for any non-adaptive schemes. Precisely, we show that the minimax rate on the budget required to achieve a target average error rate of $\varepsilon$ scales as $\Theta((m/\lambda\beta)\log(1/\varepsilon))$. The dependence on the prior $\mathscr{F}$ and $\mathscr{G}$ are solely captured in $\beta$ (the quality of the crowd as a whole) and $\lambda$ (the quality of the tasks as a whole). We show that the fundamental tradeoff for non-adaptive schemes is $\Theta((m/\lambda_{\min}\beta)\log(1/\varepsilon))$, requiring a factor of $\lambda/\lambda_{\min}$ larger budget for non-adaptive schemes. This factor of $\lambda/\lambda_{\min}$ is precisely how much we gain by adaptivity, and this gain can be made arbitrarily large in the worst case distribution $\mathscr{G}$.

## 2   Main Results

The following quantities are fundamental in capturing the dependence of the minimax rate on the distribution of task difficulties and worker reliabilities:

$$\lambda \equiv \mathbb{E}_{\mathscr{G}}\left[\frac{1}{(2q_i-1)^2}\right]^{-1}, \ \alpha \equiv \mathbb{E}_{\mathscr{G}}[(2q_i-1)^2], \text{ and } \beta \equiv \mathbb{E}_{\mathscr{F}}[(2p_j-1)^2]. \tag{2}$$

Let $n$ denote the total number of workers used, and $T_j$ denote the set of all tasks assigned to worker $j \in [n]$ and $W_i$ denote the set of all workers assigned to task $i \in [m]$ until the adaptive task assignment scheme has terminated. We consider discrete distribution $\mathscr{G}$ with $K$ types of tasks of varying difficulty levels. Define effective difficulty level of each task $i$ to be $\lambda_i \equiv (2q_i-1)^2$, and $\lambda_{\min} = \min_{i\in[m]} \lambda_i$. A task with a small $\lambda_i$ is more difficult, since $q_i$ close to 1/2 means the task is more ambiguous. Let $\delta_a$ denote fraction of total tasks having difficulty level $\lambda_a$ for $a \in [K]$ such that $\sum_{a\in[K]} \delta_a = 1$, and $\delta_{\max} \equiv \max_{a\in[K]} \delta_a$, $\delta_{\min} \equiv \min_{a\in[K]} \delta_a$.

### 2.1   Fundamental limit under the adaptive scenario

We prove a lower bound on the minimax error rate: the error that is achieved by the best inference algorithm $\hat{t}$ using the best adaptive task assignment scheme $\tau$ under a worst case worker distribution $\mathscr{F}$ and the worst-case true answers $t$ for the given distribution of difficulty level $\lambda_i$'s. Note that given $\lambda_i$, either $q_i = (1 + \sqrt{\lambda_i})/2$ in which case $t_i = 1$ or $q_i = (1 - \sqrt{\lambda_i})/2$ and $t_i = -1$. Let $\mathscr{T}_\ell$ be the set of all task assignment schemes that use at most $m\ell$ queries in total, and let $\mathscr{F}_\beta$ be the set of all the worker distributions such that expectation of worker quality is $\beta$, i.e. $\mathscr{F}_\beta \equiv \{\mathscr{F} | \mathbb{E}_{\mathscr{F}}[(2p_j-1)^2] = \beta\}$. Then we can show the following lower bound on the minimax rate on the probability of error. A proof of this theorem is provided in Section 4 in the supplementary material.

**Theorem 2.1.** *When $\beta < 1$, there exists a positive constant $C'$ such that for each task $i \in [m]$,*

$$\min_{\tau\in\mathscr{T}_\ell,\hat{t}} \ \max_{t\in\{\pm1\}^m,\mathscr{F}\in\mathscr{F}_\beta} \mathbb{P}[t_i \neq \hat{t}_i|\lambda_i] \ \geq \ \frac{1}{2}e^{-C'\lambda_i\beta\,\mathbb{E}[|W_i|\,|\,\lambda_i]}.$$

This proves a lower bound on *per task* probability of error that decays exponentially with exponent scaling as $\lambda_i \beta \mathbb{E}[|W_i| \,|\, \lambda_i]$. The easier the task ($\lambda_i$ large), the more reliable the workers are ($\beta$ large), and the more workers assigned to that task ($|W_i|$ large), the smaller the achievable error. To get a lower bound on the *average* probability of error, suppose we know the difficulties of the tasks and assign $\ell_a$ workers to tasks of difficulty $\lambda_a$. With average budget constraint $\sum_{a\in[K]} \ell_a \delta_a \leq \ell$,

$$
\min_{\tau \in \mathscr{T}_\ell, \hat{t}} \max_{t \in \{\pm 1\}^m, \mathscr{F} \in \mathscr{F}_\beta} \frac{1}{m} \sum_{i=1}^m \mathbb{P}[t_i \neq \hat{t}_i] \;\geq\; \min_{\ell_a : \sum_{a \in [K]} \delta_a \ell_a = \ell} \sum_{a=1}^K \frac{1}{2} \delta_a e^{-C' \ell_a \lambda_a \beta} \tag{3}
$$

$$
= \frac{1}{2} e^{-C' \ell \lambda \beta} \left( \sum_{a=1}^K \delta_a e^{-\lambda \sum_{a \neq a'} (\delta_{a'}/\lambda_{a'}) \log(\lambda_a/\lambda_{a'})} \right) ,
$$

where the equality follows from solving the optimization problem. Note that the summand in the bound does not depend upon the budget $\ell$, and it is lower bounded by $\delta_{\min} > 0$. The error scales as $e^{-C' \ell \lambda \beta}$, where $\lambda = 1/(\mathbb{E}[1/\lambda_i])$ as defined in (2), and captures how difficult the set of tasks are collectively. This gives a lower bound on the budget $\Gamma$ required to achieve error $\varepsilon$; there exists a constant $C''$ such that if

$$
\Gamma_\varepsilon \;\leq\; C'' \frac{m}{\lambda \beta} \log\left( \frac{\delta_{\min}}{\varepsilon} \right) , \tag{4}
$$

then no task assignment scheme (adaptive or not) with any inference algorithm can achieve error less than $\epsilon$. Intuitively, $\beta$ captures the (collective) quality of the workers as specified by $\mathscr{F}$ and $\lambda$ captures the (collective) difficulty of the tasks as specified by $\mathscr{G}$. This recovers the known fundamental limit for standard DS model where all tasks have $\lambda_i = 1$ and hence $\lambda = 1$ in [9]: $\Gamma_\varepsilon > C''' \frac{m}{\beta} \log\left(\frac{1}{\epsilon}\right)$.

## 2.2 Upper bound on the achievable error rate

We present an adaptive task assignment scheme and an iterative inference algorithm that asymptotically achieve an error rate of $C_1 e^{-(C_\delta/4)\ell \lambda \beta}$, when $m$ grows large and $\ell = \Theta(\log m)$ where $C_1 = \log_2(2\delta_{\max}/\delta_{\min}) \log_2(\lambda_1/\lambda_K)$. This matches the lower bound in (3) and the expected number of queries (or task-worker assignments) is bounded by $m\ell$. Comparing it to a fundamental lower bound in Theorem 2.1 establishes the near-optimality of our approach, and the sufficient condition to achieve average error $\varepsilon$ is for the average total budget to be larger than,

$$
\Gamma_\varepsilon \;\geq\; C' \frac{m}{\lambda \beta} \log\left( \frac{C_1}{\varepsilon} \right) . \tag{5}
$$

### 2.2.1 Adaptive algorithm

Since difficulty level is varying across the tasks, it is intuitive to assign fewer workers to easy tasks and more workers to hard tasks. Suppose we know the difficulty levels, then optimizing the lower bound (3) over $\tilde{\ell}_i$'s, it suggests to assign $\tilde{\ell}_i \simeq \ell(\lambda/\lambda_i)$ workers to the task $i$ with difficulty $\lambda_i$, when given a fixed budget of $\ell$ workers per task on average. However, the difficulty levels are not known. Non-adaptive schemes can be arbitrarily worse (see Theorem 2.4). We propose a novel approach of adaptively assigning workers in multiple rounds, refining our belief on $\lambda_i$, and making decisions on the tasks with higher confidence.

The main algorithmic component is the sub-routine in line 8-13 of Algorithm 1. For a choice of the (per task) budget $\ell_t$, we collect responses according to a $(\ell_t, r_t = \ell_t)$ regular random graph on $|M|$ tasks and $|M|$ workers. The leading eigen-vector of the non-backtracking operator on this bipartite graph, weighted by the $\pm 1$ responses reveals a noisy observation of the true class and the difficulty levels of the tasks. Let $x \in \mathbb{R}^{|M|}$ denote the top left eigenvector, computed as per Algorithm 2. Then the $i$-th entry $x_i$ asymptotically converges in the large number of tasks $m$ limit to a Gaussian random variable with mean proportional to the difficulty level $(2q_i - 1)$, with mean and variance specified in Lemma 5.1 in the the supplementary material. This non-backtracking operator approach to crowdsourcing was first introduced in [7] for the standard DS model, is a single-round non-adaptive scheme, and uses a threshold of zero to classify tasks based on the sign of $x_i$'s. We generalize their analysis to this generalized DS model in Theorem 2.3 for finite sample regime, and further give a sharper characterization based on central limit theorem in the asymptotic regime (Lemma 5.1 in the supplementary material).

This provides us a sub-routine that reveals $(2q_i - 1)$'s we want, corrupted by additive Gaussian noise. This resembles the setting in *racing algorithms* introduced in [12] where the goal is to choose the variable (i.e. task) with largest mean (i.e. easiest) with minimal budget. However, our goal is to identify the sign of the mean of the variables (i.e. classes) with sufficient accuracy. The key idea is to classify the easier tasks first with minimal budget, and then classify the remaining more difficult tasks with more budget allocated per task. We can set a threshold $\mathcal{X}_{t,u}$ at each round, and make a permanent decision on a subset of tasks that have large $x_i$'s in absolute value, since those are the tasks we are most confident about in its class, i.e. $\text{sign}(2q_i - 1)$. We are now left to choose the budget $\ell_t$ and the threshold $\mathcal{X}_{t,u}$ for each round.

We prescribe a choice using following notations. Assume that $\lambda_a$'s are indexed such that $\lambda_1 > \lambda_2 > \ldots > \lambda_K$. For simplicity, assume that $\lambda_K = \lambda_1 2^{-(T-1)}$ for some $T \in \mathbb{Z}^+ \setminus \{1\}$. Given the distribution $\{\lambda_a, \delta_a\}_{a \in [K]}$, we first bin it to get another distribution $\{\tilde{\lambda}_a, \tilde{\delta}_a\}_{a \in [T]}$ which is supported at most at $T$ points. We take $\tilde{\lambda}_1 = \lambda_1$ and $\tilde{\lambda}_{a+1} = \tilde{\lambda}_a 2^{-1}$ for each $a \in [T-1]$. $\tilde{\delta}_a$ is the total fraction of tasks whose difficulty $\lambda_i$ is smaller than $\lambda_1 2^{-(a-2)}$ and larger than $\lambda_1 2^{-(a-1)}$. Precisely, $\tilde{\delta}_a = \sum_{a' \in [K]} \delta_{a'} \mathbb{I}\{\lambda_1 / 2^{(a-1)} \le \lambda_{a'} < \lambda_1 / 2^{(a-2)}\}$, for $a \in [T]$. The choice of 2 for the ratio of $\tilde{\lambda}_a$'s is arbitrary and can be further optimized for a given distribution of $\lambda_i$'s. For ease of notations in writing the algorithm, we re-index the binned distribution to get $\{\tilde{\lambda}_a, \tilde{\delta}_a\}_{a \in [\tilde{T}]}$, for $\tilde{T} \le T$, such that $\tilde{\delta}_a \ne 0$ for all $a \in \tilde{T}$. Note that $\tilde{T} \le \lceil \log_2(\lambda_1 / \lambda_K) \rceil$.

We start with a set of all tasks $M = [m]$. A fraction of tasks are classified in each round and the un-classified ones are taken to the next round. At round $t \in \{1, \ldots, \tilde{T}\}$, our goal is to classify sufficient fraction of those tasks in the same difficulty group $\{i \in M : \lambda_i = \tilde{\lambda}_t\}$ to be classified with desired level of accuracy. If $\ell_t$ is too low and/or threshold $\mathcal{X}_{t,u}$ too small, then misclassification rate will be too large. If $\ell_t$ is too large, we are wasting our budget unnecessarily. If $\mathcal{X}_{t,u}$ is too large, not enough tasks will be classified. We choose $\ell_t = \ell C_\delta \tilde{\lambda} / \tilde{\lambda}_t$ and an appropriate $\mathcal{X}_{t,u}$ to ensure that the misclassification probability is at most $C_1 e^{-(C_\delta/4)\lambda \beta \ell}$ based on the central limit theorem on the leading eigen vector (see (21) in the supplementary material). We run this sub-routine $s_t = \max\{0, \lceil \log_2(\tilde{\delta}_t(1 + \gamma_t) / \tilde{\delta}_{t+1} \gamma_{t+1}) \rceil\}$ times to ensure that enough fraction from $t$-th group is classified. We make sure that the expected number of unclassified tasks is at most equal to the number of tasks in the next group, i.e., difficulty level $\lambda_i = \tilde{\lambda}_{t+1}$. We provide a near-optimal performance guarantee for $\gamma_t = 1$ for all $t \in [\tilde{T}]$, and $\gamma_t$ provides an extra degree of freedom for practitioners to further optimize the efficiency.

Note that statistically, the fraction of the $t$-th group (i.e. tasks with difficulty $\tilde{\lambda}_t$) that get classified before the $t$-th round is very small as the threshold set in these rounds is more than their absolute mean message. Most tasks with $\tilde{\lambda}_t$ will get classified in round $t$. Further, the binning of the original given distribution to get $\{\tilde{\lambda}_a, \tilde{\delta}_a\}$ ensures that $\ell_{t+1} \ge 2\ell_t$. It ensures that the total extraneous budget spent on $\tilde{\lambda}_t$ tasks is not more than a constant times the allocated budget of those tasks, and the constant can be made one, by changing the initial choice of $\ell_1$ by a constant factor.

### 2.2.2 Performance Guarantee

Since we are not wasting any budget on any of the tasks, with the right choice of the constant $C_\delta$, we are guaranteed that this algorithm uses at most $m\ell$ assignments in expectation. One caveat is that, the threshold $\mathcal{X}_{t,u}$ depends on $\alpha_{t,u} = (1/|M|) \sum_{i \in [M]} \lambda_i$, which is the average difficulty of the remaining tasks. As the remaining tasks are changing over the course of the algorithm, we need to estimate this value in each sub-routine. We provide an estimator of $\alpha_{t,u}$ in Algorithm 3 (in the supplementary material) that only uses the sampled responses that are already collected. All numerical results are based on this estimator. However, analyzing the sensitivity of the performance with respect to the estimation error in $\alpha_{t,u}$ is quite challenging, and for a theoretical analysis, we assume we have access to an oracle that provides the exact value of $\alpha_{t,u}$, replacing Algorithm 3.

**Theorem 2.2.** *Suppose Algorithm 3 returns the exact value of $\alpha_{t,u} = (1/|M|) \sum_{i \in [M]} \lambda_i$. With the choice of $\gamma_a = 1$ for all $a \in [\tilde{T}]$ and $C_\delta = (4 + \lceil \log(2\delta_{\max}/\delta_{\min}) \rceil)^{-1}$ for any given distribution of task difficulty $\{\lambda_a, \delta_a\}_{a \in [K]}$ of $m$ tasks and an average number of workers per task $\ell = \Theta(\log m)$, the expected number of queries made by Algorithm 1 is asymptotically bounded by $\lim_{m \to \infty} \sum_{t \in [\tilde{T}], u \in s_t} \ell_t \mathbb{E}[|M_{t,u}|] / (m\ell) \le 1$, where $M_{t,u}$ is the number of tasks remaining at round*

$(t, u)$. *Further, Algorithm 1 returns estimates $\{\hat{t}_i\}_{i \in [m]}$ that asymptotically achieves,*

$$\lim_{m \to \infty} \frac{1}{m} \sum_{i=1}^{m} \mathbb{P}[t_i \neq \hat{t}_i] \quad \leq \quad C_1 e^{-(C_\delta/4)\ell \lambda \beta} , \tag{6}$$

*where $C_1 = \log_2(2\delta_{\max}/\delta_{\min}) \log_2(\lambda_1/\lambda_K)$ for $\lambda\beta$ scaling as $1/\ell$ such that $\ell\lambda\beta = \Theta(1)$.*

A proof of this theorem is provided in Section 5 in the supplementary material. This shows the near-optimal sufficient condition of our approach in (5). The constant $C_\delta$ can be improved by optimizing over the choice of $\gamma_a$'s by minimizing the expected number of queries that the algorithm makes.

---

**Algorithm 1** Adaptive Task Assignment and Inference Algorithm

---

**Require:** $m, \{\tilde{\lambda}_a, \tilde{\delta}_a\}_{a \in [\tilde{T}]}, \ell, C_\delta, \{\gamma_a\}_{a \in [\tilde{T}]}, \alpha, \beta, \mu = \mathbb{E}[2p_j - 1]$
**Ensure:** Estimate $\{\hat{t}_i\}_{i \in [m]}$

1: $M \leftarrow \{1, 2, \cdots, m\}, \tilde{\lambda} = \left( \sum_{a \in [\tilde{T}]} (\tilde{\delta}_a/\tilde{\lambda}_a) \right)^{-1}$
2: **for all** $t = 1, 2, \cdots, \tilde{T}$ **do**
3:      $\ell_t \leftarrow (\ell C_\delta \tilde{\lambda})/\tilde{\lambda}_t, r_t \leftarrow \ell_t$
4:      $s_t \leftarrow \max \left\{0, \left\lceil \log \left( \frac{\tilde{\delta}_t(1+\gamma_t)}{\tilde{\delta}_{t+1}\gamma_{t+1}} \right) \right\rceil \right\} \mathbb{I}\{t < \tilde{T}\} + 1 \, \mathbb{I}\{t = \tilde{T}\}$
5:      **for all** $u = 1, 2, \cdots, s_t$ **do**
6:        **if** $M \neq \varnothing$ **then**
7:          $n \leftarrow |M| , k \leftarrow \sqrt{\log |M|}$
8:          Draw $E \in \{0, 1\}^{|M| \times n} \sim (\ell_t, r_t)$-regular random graph
9:          Collect answers $\{A_{i,j} \in \{1, -1\}\}_{(i,j) \in E}$
10:         $\{x_i\}_{i \in M} \leftarrow$ Algorithm 2 $\left[ E, \{A_{i,j}\}_{(i,j) \in E}, k \right]$
11:         $\alpha_{t,u} \leftarrow$ Algorithm 3 $\left[ E, \{A_{i,j}\}_{(i,j) \in E}, \ell_t, r_t \right]$
12:         $\mathcal{X}_{t,u} \leftarrow \sqrt{\tilde{\lambda}_t} \mu \ell_t \left( (\ell_t - 1)(r_t - 1)\alpha_{t,u}\beta \right)^{k-1} \mathbb{I}\{t < \tilde{T}\} + 0 \, \mathbb{I}\{t = \tilde{T}\}$
13:         $\{\hat{t}_i = \mathbb{I}\{x_i > \mathcal{X}_{t,u}\} - \mathbb{I}\{x_i < -\mathcal{X}_{t,u}\}\}_{i \in M}, M \leftarrow \{i \in M : |x_i| \leq \mathcal{X}_{t,u}\}$
14:        **end if**
15:      **end for**
16: **end for**

---

**Algorithm 2** Message-Passing Algorithm

---

**Require:** $E \in \{0, 1\}^{|M| \times n}, \{A_{ij} \in \{1, -1\}\}_{(i,j) \in E}, k_{\max}$
**Ensure:** $\{x_i \in \mathbb{R}\}_{i \in [|M|]}$

1: **for all** $(i, j) \in E$ **do**
2:      Initialize $y_{j \to i}^{(0)}$ with random $Z_{j \to i} \sim \mathcal{N}(1, 1)$
3: **end for**
4: **for all** $k = 1, 2, \cdots, k_{\max}$ **do**
5:      **for all** $(i, j) \in E$ **do**
6:        $x_{i \to j}^{(k)} \leftarrow \sum_{j' \in W_i \backslash j} A_{ij'} y_{j' \to i}^{k-1}$
7:      **end for**
8:      **for all** $(i, j) \in E$ **do**
9:        $y_{j \to i}^{(k)} \leftarrow \sum_{i' \in T_j \backslash i} A_{i'j} x_{i' \to j}^k$
10:     **end for**
11: **end for**
12: **for all** $i \in [m]$ **do**
13:      $x_i \leftarrow \sum_{j \in W_i} A_{ij} y_{j \to i}^{k_{\max}-1}$
14: **end for**

---

In Figure 1, we compare performance of our algorithm with majority voting and also non-adaptive version of our Algorithm 1, where we assign to each task $\ell$ (the given budget) number of workers in

one round and set classification threshold $\mathcal{X}_{t,u} = 0$ so as to classify all the tasks. This non-adaptive special case has been introduced for the standard DS model in [9].

We make a slight modification to Algorithm 1. In the final round, when the classification threshold is set to zero, we include all the responses collected thus far when running the message passing Algorithm 2, and not just the fresh samples collected in that round. This creates dependencies between rounds, which makes the analysis challenging. However, in practice we see improved performance and it allows us to use the given fixed budget efficiently.

We run synthetic experiments with $m = 1800$ and fix $n = 1800$ for the non-adaptive version. The crowds are generated from the spammer-hammer model with hammer probability equal to 0.3. In the left panel, we take difficulty level $\lambda_a$ to be uniformly distributed over $\{1, 1/4, 1/16\}$, that gives $\lambda = 1/7$. In the right panel, we take $\lambda_a = 1$ with probability $3/4$, otherwise we take it to be $1/4$ or $1/16$ with equal probability, that gives $\lambda = 4/13$. As predicted from the theoretical analysis, our adaptive algorithm improves significantly over its non-adaptive version. In particular, for the left panel, the non-adaptive algorithm's error scaling depends on smallest $\lambda_i$ that is $1/16$ while for the adaptive algorithm it scales with $\lambda = 1/7$. In the figure, it can be seen that the adaptive algorithm requires approximately $(7/16)\ell$ queries to acheive the same error as achieved by the non-adaptive one using $\ell$ queries. This gap widens in the right panel to approximately $(13/64)$ as predicted, and the adaptive algorithm achieves zero error as the number of queries increase. For a fair comparison with the non-adaptive version, we fix total budget to be $m\ell$ and assign workers in each round until the budget is exhausted. $C_\delta$ is 1 and $s_t = 1$ for $t \in \{1, 2, 3\}$.

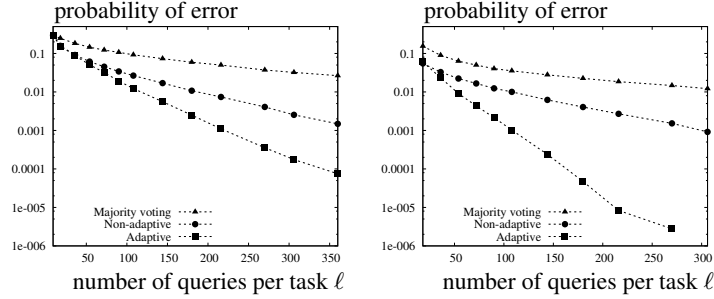

Figure 1: Algorithm 1 improves significantly over its non-adaptive version and majority voting.

## 2.3 Achievable error rate under the non-adaptive scenario

Consider a non-adaptive version of our approach where we apply it for one round using an $(\ell, r)$ random regular graph, where $\ell$ is the given budget. Naturally, the classification threshold is set to $\mathcal{X}_{t,u} = 0$ so as to classify all the tasks. We provide a sharp upper bound on the achieved error, that holds for all (non-asymptotic) regimes of $m$. Define $\sigma_k^2$ as

$$\sigma_k^2 \;\equiv\; \frac{2\beta}{\mu^2 \big(\hat{\ell}\hat{r}(\alpha\beta)^2\big)^{k-1}} + 3\left(1 + \frac{1}{\hat{r}\alpha\beta}\right) \frac{1 - 1/\big(\hat{\ell}\hat{r}(\alpha\beta)^2\big)^{k-1}}{1 - 1/\big(\hat{\ell}\hat{r}(\alpha\beta)^2\big)}. \tag{7}$$

This captures the effective variance in the sub-Gaussian tail of the messages $x_i$'s after $k$ iterations of the inference algorithm (Algorithm 2), as shown in the proof of the following theorem (see the supplementary material in Section 6).

**Theorem 2.3.** *For any $\ell > 1$ and $r > 1$, suppose $m$ tasks are assigned according to a random $(\ell, r)$-regular graph drawn from the configuration model. If $\mu > 0$, $\hat{\ell}\hat{r}\alpha^2\beta^2 > 1$, and $\hat{r}\alpha > 1$, then for any $t \in \{\pm 1\}^m$, the estimate $\hat{t}_i = \mathrm{sign}(x_i)$ after $k$ iterations of Algorithm 2 achieves*

$$\mathbb{P}\big[t_i \neq \hat{t}_i^{(k)}\big|\lambda_i\big] \;\leq\; e^{-\ell\beta\lambda_i/(2\sigma_k^2)} + \frac{3\ell r}{m}(\hat{\ell}\hat{r})^{2k-2}. \tag{8}$$

Therefore, the average error rate is bounded by

$$\frac{1}{m}\sum_{i=1}^{m}\mathbb{P}[t_i \neq \hat{t}_i^{(k)}] \;\leq\; \mathbb{E}_{\mathscr{G}}\left[e^{\frac{-\ell\beta\lambda_i}{2\sigma_k^2}}\right] + \frac{3\ell r}{m}(\hat{\ell}\hat{r})^{2k-2}. \tag{9}$$

The second term, which is the probability that the resulting $(\ell, r)$ regular random graph is not locally tree-like, can be made small for large $m$ as long as $k = O(\sqrt{\log m})$ (which is the choice we make in Algorithm 1). Hence, the dominant term in the error bound is the first term. Further, when we run our algorithm for large enough numbers of iterations, $\sigma_k^2$ converges linearly to a finite limit $\sigma_\infty^2 \equiv \lim_{k \to \infty} \sigma_k^2$ such that $\sigma_\infty^2 = 3(1 + 1/(\hat{r}\alpha\beta))(\hat{\ell}\hat{r}\alpha\beta)^2/((\hat{\ell}\hat{r}\alpha\beta)^2 - 1)$, which for large enough $\hat{r}\alpha\beta$ and $\hat{\ell}\hat{r}$ is upper bounded by a constant. Hence, for a wide range of parameters, the average error in (9) is dominated by $\mathbb{E}_{\mathscr{G}}\left[e^{-\ell\beta\lambda_i/2\sigma_k^2}\right] = \sum_a \delta_a e^{-C\ell\beta\lambda_a}$. When all $\delta$'s are strictly positive, the error is dominated by the difficult tasks with $\lambda_{\min} = \min_a \lambda_a$, as illustrated in Figure 2. Hence, it is sufficient to have budget $\Gamma_\varepsilon \geq C''m/(\lambda_{\min}\beta)\log(1/\varepsilon)$ to achieve an average error of $\varepsilon > 0$. Such a scaling is also necessary as we show in the next section.

This is further illustrated in Figure 2. The error decays exponentially in $\ell$ and $\beta$ as predicted, but the rate of decay crucially hinges on the difficulty level. We run synthetic experiments with $m = n = 1000$ and the crowds are generated from the spammer-hammer model where $p_j = 1$ with probability $\beta$ and $1/2$ otherwise. We fix $\beta = 0.3$ and vary $\ell$ in the left figure and fix $\ell = 30$ and vary $\beta$ in the right figure. We let $q_i$'s take values in $\{0.6, 0.8, 1\}$ with equal probability such that $\alpha = 1.4/3$. The error rate of each task grouped by their difficulty is plotted in the dashed lines, matching predicted $e^{-\Omega(\ell\beta(2q_i - 1)^2)}$. The average error rates in solid lines are dominated by those of the difficult tasks, which is a universal drawback for all non-adaptive schemes.

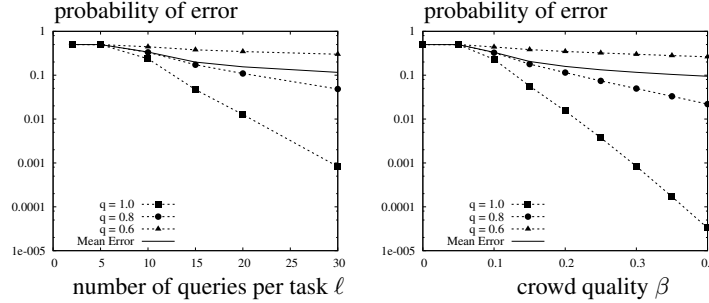

Figure 2: Non-adaptive schemes suffer as average error is dominated by difficult tasks.

## 2.4 Fundamental limit under the non-adaptive scenario

Theorem 2.3 implies that it suffices to assign $\ell \geq (c/(\beta\lambda_i))\log(1/\varepsilon)$ to achieve an error smaller than $\varepsilon$ for a task $i$. We show in the following theorem that this scaling is also necessary. Hence, applying one round of Algorithm 1 is near-optimal in the non-adaptive scenario compared to a minimax rate where the nature chooses the worst distribution of worker $p_j$'s among the set of distributions with the same $\beta$. We provide a proof of the theorem in Section 7 in the supplementary material.

**Theorem 2.4.** *There exists a positive constant $C'$ and a distribution $\mathscr{F}$ of workers with average reliability $\mathbb{E}[(2p_j - 1)^2] = \beta$ s.t. when $\lambda_i < 1$, if the number of workers assigned to task $i$ by any non-adaptive task assignment scheme is less than $(C'/(\beta\lambda_i))\log(1/\epsilon)$, then no algorithm can achieve conditional probability of error on task $i$ less than $\epsilon$ for any $m$ and $r$.*

Since in this non-adaptive scheme, task assignments are done a priori, there are on average $\ell$ workers assigned to any set of tasks of the same difficulty. Hence, if the total budget is less than

$$\Gamma_\varepsilon \leq C'\frac{m}{\lambda_{\min}\beta}\log\frac{\delta_{\min}}{\varepsilon}, \tag{10}$$

then no algorithm can achieve average error less than $\varepsilon$, where $\lambda_{\min} = \min_a \lambda_a$. Compared to the adaptive case in (4) (nearly achieved in (5)), the gain of adaptivity is a factor of $\lambda/\lambda_{\min}$. The RHS is negative when $\delta_{\min} < \varepsilon$, and can be tightened to $C'(m/\lambda_a\beta)\log(\sum_{b=1}^a \delta_b/\varepsilon)$ where $a$ is the smallest integer such that $\sum_{b=1}^a \delta_b > \varepsilon$.

## Acknowledgements

This work is supported by NSF SaTC award CNS-1527754, and NSF CISE award CCF-1553452.

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
