[Supplementary Material]

# Supplementary Material for "Achieving Budget-optimality with Adaptive Schemes in Crowdsourcing"

## 3 Algorithm for parameter estimation

In this section we present a spectral algorithm for estimating $\alpha$, to be used in the inner-loop of Algorithm 1. Consider the data matrix $\widetilde{A}$ defined below. A simple analysis shows that $\mathbb{E}[\widetilde{A}]$ is a rank one matrix with $\|\mathbb{E}[\widetilde{A}]\| = \sqrt{\ell r \alpha \beta}$. A typical random matrix analysis shows that spectral norm of the noise matrix $\|\widetilde{A} - \mathbb{E}[\widetilde{A}]\|$ is upper bounded by $C(\ell r)^{1/4}$ for some constant $C$. Therefore, in the limit of $|M|$, since $\ell = r = \Theta(\log|M|)$, we have $\|\widetilde{A}\|/\sqrt{\ell r \beta} = \|\mathbb{E}[\widetilde{A}]\|/\sqrt{\ell r \beta} + o(1)$. Hence, $\alpha$ is close to the squared normalized top singular value of $\widetilde{A}$.

---

**Algorithm 3** Parameter Estimation Algorithm

---

**Require:** $E \in \{0,1\}^{|M| \times n}$, $\{A_{ij}\}_{(i,j) \in E}$, $\ell, r, \beta$
**Ensure:** $\alpha$
  1: Construct matrix $\widetilde{A} \in \{0, \pm 1\}^{|M| \times n}$ such that

$$\widetilde{A}_{i,j} = \left\{ \begin{array}{rl} A_{i,j} & \text{, if } (i,j) \in E \\ 0 & \text{, otherwise} \end{array} \right.$$

     for all $i \in [|M|]$, $j \in [n]$.
  2: Set $\sigma_1(\widetilde{A})$ to be the top singular value of matrix $\widetilde{A}$
  3: $\alpha \leftarrow \left( \sigma_1(\widetilde{A})/\sqrt{\ell r \beta} \right)^2$

---

## 4 Proof of Theorem 2.1

We will show that there exists a family of worker reliability distributions $\mathscr{F}$ such that for any adaptive task assignment scheme that assigns $\mathbb{E}[|W_i||q_i]$ workers in expectation to a task $i$ conditioned on its difficulty $q_i$, the conditional probability of error of task $i$ conditioned on $q_i$ is lower bounded by $\exp\left(-C'\lambda_i \beta \mathbb{E}[|W_i||q_i]\right)$. We define the following family of distributions according to the spammer-hammer model with imperfect hammers. We assume that $\beta < a^2$ and

$$p_j = \left\{ \begin{array}{rl} 1/2, & \text{w.p. } 1 - \beta/a^2, \\ 1/2(1+a), & \text{w.p. } \beta/a^2. \end{array} \right. ,$$

such that $E[(2p_j - 1)^2] = \beta$. Let $\mathbb{E}[W_i|q_i]$ denote the expected number of workers conditioned on the task difficulty $q_i$, that the adaptive task assignment scheme assigns to the task $i$. We consider a labeling algorithm that has access to an oracle that knows reliability of every worker (all the $p_j$'s). Focusing on a single task $i$, since we know who the spammers are and spammers give no information about the task, we only need the responses from the reliable workers in order to make an optimal estimate. Let $\mathscr{E}_i$ denote the conditional error probability of the optimal estimate conditioned on the realizations of the answers $\{A_{ij}\}_{j \in W_i}$ and the worker reliability $\{p_j\}_{j \in W_i}$. We have $\mathbb{E}[\mathscr{E}_i|q_i] \equiv \mathbb{P}[t_i \neq \hat{t}_i|q_i]$. The following lower bound on the error only depends on the number of reliable workers, which we denote by $\ell_i$.

Without loss of generality, let $t_i = +1$. Then, if all the reliable workers agreed on "–" answers, the maximum likelihood estimation would be "–" for this task, and vice-versa. For a fixed number of $\ell_i$ responses, the probability of error is minimum when all the workers agreed. Therefore, since probability that a worker gives "–" answer is $p_j(1 - q_i) + q_i(1 - p_j) = (1 - a(2q_i - 1))/2$ from (1), we have,

$$\mathbb{E}[\mathscr{E}_i|q_i, \ell_i] \geq \mathbb{E}[\mathscr{E}_i|\text{all } \ell_i \text{ reliable workers agreed}, q_i, \ell_i] \geq \frac{1}{2}\left(\frac{1 - a(2q_i - 1)}{2}\right)^{\ell_i}, \quad (11)$$

for any realizations of $\{A_{ij}\}$ and $\{p_j\}$. By convexity and Jensen's inequality, it follows that

$$\mathbb{E}[\ell_i|q_i] \geq \frac{\log(2\mathbb{E}[\mathscr{E}_i|q_i])}{\log((1 - a(2q_i - 1))/2)}. \quad (12)$$

When we recruit $|W_i|$ workers, using Doob's Optional-Stopping Theorem [20, 10.10], conditional expectation of reliable number of workers is

$$\mathbb{E}[\ell_i | q_i] = (\beta/a^2)\mathbb{E}[|W_i| \, | q_i] \,. \tag{13}$$

Therefore, from (12) and (13), we get

$$\mathbb{E}[|W_i| \, | q_i] \geq \frac{1}{\beta} \frac{a^2}{\log((1 - a(2q_i - 1))/2)} \log(2\mathbb{E}[\mathscr{E}_i | q_i]) \,. \tag{14}$$

Maximizing over all choices of $a \in (0, 1)$, we get,

$$\mathbb{E}[|W_i| \, | q_i] \geq \frac{-0.27}{\beta(2q_i - 1)^2} \log(2\mathbb{E}[\mathscr{E}_i | q_i]) \,, \tag{15}$$

for $a = 0.8/(2q_i - 1)$ which as per our assumption of $\beta < a^2$ requires that $\beta(2q_i - 1)^2 < 0.64$. By changing the constant in the bound, we can ensure that the bound holds for any value of $\beta$ and $q_i$. Theorem readily follows from Equation (15).

To verify Equation 13: Define $X_{i,k}$ for $k \in [|W_i|]$ to be a Bernoulli random variable, for a fixed $i \in [m]$ and fixed task difficulty $q_i$. Let $X_{i,k}$ take value one when the $k$-th recruited worker for task $i$ is reliable and zero otherwise. Observe that the number of reliable workers is $\ell_i = \sum_{k=1}^{|W_i|} X_{i,k}$. From the spammer-hammer model that we have considered, $\mathbb{E}[X_{k,i} - \beta/a^2] = 0$. Define $Z_{i,k} \equiv \sum_{k'=1}^{k}(X_{i,k'} - \beta/a^2)$ for $k \in [|W_i|]$. Since $\{(X_{i,k} - \beta/a^2)\}_{k \in [|W_i|]}$ are mean zero i.i.d. random variables, $\{Z_{i,k}\}_{k \in [|W_i|]}$ is a martingale with respect to the filtration $\mathcal{F}_{i,k} = \sigma(X_{i,1}, X_{i,2}, \cdots, X_{i,k})$. Further, it is easy to check that the random variable $|W_i|$ for a fixed $q_i$ is a stopping time with respect to the same filtration $\mathcal{F}_{i,k}$ and is almost surely bounded assuming the budget is finite. Therefore using Doob's Optional-Stopping Theorem [20, 10.10], we have $\mathbb{E}[Z_{i,|W_i|}] = \mathbb{E}[Z_{i,1}] = 0$. That is we have, $\mathbb{E}[X_{i,1} + X_{i,2} + \cdots + X_{i,|W_i|}] = (\beta/a^2)\mathbb{E}[|W_i|]$. Since this is true for any fixed task difficulty $q_i$, we get Equation (13).

## 5 Proof of Theorem 2.2

First we show that the messages returned by Algorithm 2 are normally distributed and identify their conditional mean and conditional variance in the following lemma. Assume the number of tasks is $m$, the number of workers used is $n$, and the task assignment is performed according to $(\ell, r)$ regular random graph. To simplify the notation, let $\hat{\ell} \equiv \ell - 1$, $\hat{r} \equiv r - 1$, and define $\mu \equiv \mathbb{E}[2p_j - 1]$. When $\ell$ and $r$ are increasing with the problem size, the messages converge to a Gaussian distribution due to the central limit theorem. We provide a proof of this lemma in Section 5.1.

**Lemma 5.1.** *Suppose for $\ell = \Theta(\log m)$ and $r = \Theta(\log m)$, tasks are assigned according to $(\ell, r)$-regular random graphs. In the limit $m \to \infty$, if $\mu > 0$, then after $k = \Theta(\sqrt{\log m})$ number of iterations in Algorithm 2, the conditional mean $\mu_q^{(k)}$ and the conditional variance $\left(\rho_q^{(k)}\right)^2$ conditioned on the task difficulty q of the message $x_i$ are*

$$\mu_q^{(k)} = (2q - 1)\mu\ell(\hat{\ell}\hat{r}\alpha\beta)^{(k-1)} \,,$$

$$\left(\rho_q^{(k)}\right)^2 = \mu^2\ell(\hat{\ell}\hat{r}\alpha\beta)^{2(k-1)}\left(\alpha - (2q-1)^2 + \frac{\alpha\hat{\ell}(1 - \alpha\beta)(1 + \hat{r}\alpha\beta)\left(1 - (\hat{\ell}\hat{r}\alpha^2\beta^2)^{-(k-1)}\right)}{\hat{\ell}\hat{r}\alpha^2\beta^2 - 1}\right)$$
$$+ \ell(2 - \mu^2\alpha)(\hat{\ell}\hat{r})^{k-1} \,. \tag{16}$$

We will show that the probability of misclassification for any task in any round in Algorithm 1 is upper bounded by $e^{-(C_\delta/4)\ell\lambda\beta}$ and the expected total number of worker assignments is at most $m\ell$ when $\{\gamma_a = 1\}_{a \in [\tilde{T}]}$. Since, there are utmost $C_1 = s_{\max}\tilde{T} = \log_2(2\delta_{\max}/\delta_{\min})\log_2(\lambda_1/\lambda_K)$ rounds, using union bound we get the desired probability of error.

Let's consider any task $i \in [m]$ having difficulty $\lambda_i$. Without loss of generality assume that $t_i = 1$ that is $q_i > 1/2$. Let us assume that the task $i$ gets classified in the $(t, u)$-round, $t \in [\tilde{T}]$, $u \in [s_t]$. That is the number of workers assigned to the task $i$ when it gets classified is $\ell_t = (\ell C_\delta\tilde{\lambda})/\tilde{\lambda}_t$ and the threshold $\mathcal{X}_{t,u}$ set in that round for classification is $\mathcal{X}_{t,u} = \sqrt{\tilde{\lambda}_t}\mu\ell_t\left((\ell_t - 1)(r_t - 1)\alpha_{t,u}\beta\right)^{k_t-1}$.

From Lemma 5.1 the message $x_i$ returned by Algorithm 2 is Gaussian with conditional mean and conditional variance as given in (16). Therefore in the limit of $m$, the probability of error in task $i$ is

$$\lim_{m\to\infty} \mathbb{P}\big[\hat{t}_i \neq t_i | q_i\big] = \lim_{m\to\infty} \mathbb{P}\big[x_i < -\mathcal{X}_{t,u} | q_i\big]$$

$$= \lim_{m\to\infty} Q\Big(\frac{\mu_{q_i}^{(k)} + \mathcal{X}_{t,u}}{\rho_{q_i}^{(k)}}\Big) \tag{17}$$

$$\leq \lim_{m\to\infty} \exp\Big(\frac{-(\mu_{q_i}^{(k)} + \mathcal{X}_{t,u})^2}{2(\rho_{q_i}^{(k)})^2}\Big) \tag{18}$$

$$= \exp\Big(\frac{-((2q_i - 1) + \sqrt{\tilde{\lambda}_t})^2 \ell_t \beta}{2(1 - (2q_i - 1)^2 \beta)}\Big) \tag{19}$$

$$\leq \exp\Big(\frac{-\tilde{\lambda}_t \ell_t \beta}{2}\Big)$$

$$= \exp\Big(\frac{-C_\delta \tilde{\lambda} \ell \beta}{2}\Big) \tag{20}$$

$$\leq \exp\Big(\frac{-C_\delta \lambda \ell \beta}{4}\Big), \tag{21}$$

where $Q(\cdot)$ in (17) is the tail probability of a standard Gaussian distribution, and (18) uses the Chernoff bound. (19) follows from substituting conditional mean and conditional variance from Equation (16), and using $\ell_t = \Theta(\log m)$, $k = \Theta(\sqrt{\log m})$ where $m$ grows to infinity. (20) uses $\ell_t = (\ell C_\delta \tilde{\lambda})/\tilde{\lambda}_t$ and (21) uses the fact that for the binned distribution $\{\tilde{\lambda}_a, \tilde{\delta}_a\}_{a \in [\tilde{T}]}$, $\tilde{\lambda} = \big(\sum_{a \in [\tilde{T}]} (\tilde{\delta}_a / \tilde{\lambda}_a)\big)^{-1} \geq \lambda/2$. We have established that our approach guarantees the desired level of accuracy. We are left to show that we use utmost $m\ell$ assignments in expectation.

We upper bound the expected total number of workers used for tasks of difficulty level $\tilde{\lambda}_a$'s for each $1 \leq a \leq \tilde{T}$. Recall that our adaptive algorithm runs in $\tilde{T}$ rounds indexed by $t$, where each round $t$ further runs $s_t$ sub-rounds. The total expected number of workers assigned to $\tilde{\delta}_a$ fraction of tasks of difficulty $\tilde{\lambda}_a$ in $t = 1$ to $t = a - 1$ rounds is upper bounded by $m\tilde{\delta}_a \sum_{t=1}^{a-1} s_t \ell_t$. The upper bound assumes the worst-case (in terms of the budget) that these tasks do not get classified in any of these rounds as the threshold $\mathcal{X}$ set in these rounds is more than absolute value of the conditional mean message $x$ of these tasks.

Next, in $s_{t=a}$ sub-rounds the threshold $\mathcal{X}$ is set less than or equal to the absolute value of the conditional mean message $x$ of these tasks, i.e. $\mathcal{X} \leq |\mu_{q_a}^{(k)}|$ for $(2q_a - 1)^2 = \tilde{\lambda}_a$. Therefore, in each of these $s_a$ sub-rounds, probability of classification of these tasks is at least $1/2$. That is the expected total number of workers assigned to these tasks in $s_a$ sub-rounds is upper bounded by $2m\tilde{\delta}_a \ell_a$. Further, $s_a$ is chosen such that the fraction of these tasks remaining un-classified at the end of $s_a$ sub-rounds is utmost same as the fraction of the tasks having difficulty $\tilde{\lambda}_{a+1}$. That is to get the upper bound, we can assume that the fraction of $\tilde{\lambda}_{a+1}$ difficulty tasks at the start of $s_{a+1}$ sub-rounds is $2\tilde{\delta}_{a+1}$, and the fraction of $\tilde{\lambda}_a$ difficulty tasks at the start of $s_{a+1}$ sub-rounds is zero. Further, recall that we have set $s_{\tilde{T}} = 1$ as in this round our threshold $\mathcal{X}$ is equal to zero. Therefore, we have the following upper bound on the expected total number of worker assignments.

$$\sum_{i=1}^m \mathbb{E}[|W_i|] \leq 2m\tilde{\delta}_1 \ell_1 + \sum_{a=2}^{\tilde{T}-1} 4m\tilde{\delta}_a \ell_a + 2m\tilde{\delta}_{\tilde{T}} \ell_{\tilde{T}} + \sum_{a=2}^{\tilde{T}} \Big(m\tilde{\delta}_a \sum_{b=1}^{a-1} s_b \ell_b\Big)$$

$$\leq \sum_{a=1}^{\tilde{T}} 4m\tilde{\delta}_a \ell_a + s_{\max} \sum_{a=1}^{\tilde{T}} m\tilde{\delta}_a \ell_a \tag{22}$$

$$\leq (4 + \lceil \log(2\delta_{\max}/\delta_{\min}) \rceil) \sum_{a=1}^{\tilde{T}} m\tilde{\delta}_a \ell_a \tag{23}$$

$$\leq (4 + \lceil \log(2\delta_{\max}/\delta_{\min}) \rceil) m\ell C_\delta \tag{24}$$

$$= m\ell, \tag{25}$$

Equation (22) uses the fact that $\ell_t = (\ell C_\delta \tilde{\lambda})/\tilde{\lambda}_t$ where $\tilde{\lambda}_t$'s are separated apart by at least a ratio of 2 (recall the binning distribution), therefore $\sum_{t=1}^{a-1} \ell_t \le \ell_a$. Equation (23) follows from the choice of $s_t$'s in the algorithm. Equation (24) follows from using $\ell_t = (\ell C_\delta \tilde{\lambda})/\tilde{\lambda}_t$ and $\tilde{\lambda} = (\sum_{a \in [\tilde{T}]} (\tilde{\delta}_a/\tilde{\lambda}_a))^{-1}$, and Equation (25) uses $C_\delta = (4 + \lceil \log(2\delta_{\max}/\delta_{\min}) \rceil)^{-1}$.

## 5.1 Proof of Lemma 5.1

We will prove it for a randomly chosen node $\mathbf{I}$, and all the analyses naturally holds for a specific $i$, when conditioned on $q_i$. In our algorithm, we perform task assignment on a random bipartite graph $\mathbf{G}([m] \cup [n], E)$ constructed according to the configuration model. Let $\mathbf{G}_{i,k}$ denote a subgraph of $\mathbf{G}([m] \cup [n], E)$ that includes all the nodes that are within $k$ distance from the the "root" $i$. If we run our inference algorithm for one run to estimate $\hat{t}_i$, we only use the responses provided by the workers who were assigned to task $i$. That is we are running inference algorithm only on the local neighborhood graph $\mathbf{G}_{i,1}$. Similarly, when we run our algorithm for $k$ iterations to estimate $\hat{t}_i$, we perform inference only on the local subgraph $\mathbf{G}_{i,2k-1}$. Since we update both task and worker messages at each iteration, the local subgraph grows by distance two at each iteration. We use a result from [9] to show that the local neighborhood of a randomly chosen task node $\mathbf{I}$ is a tree with high probability. Therefore, assuming that the graph is locally tree like with high probability, we can apply a technique known as *density evolution* to estimate the conditional mean and conditional variance. The next lemma shows that the local subgraph converges to a tree in probability, in the limit $m \to \infty$ for the specified choice of $\ell, r$ and $k$.

**Lemma 5.2** (Lemma 5 from [9]). *For a random $(\ell, r)$-regular bipartite graph generated according to the configuration model,*

$$\mathbb{P}\big[\mathbf{G}_{\mathbf{I},2k-1} \text{ is not a tree}\big] \le \big((\ell-1)(r-1)\big)^{2k-2} \frac{3\ell r}{m}. \tag{26}$$

**Density Evolution.** Let $\{x_{i \to j}^{(k)}\}_{(i,j) \in E}$ and $\{y_{j \to i}^{(k)}\}_{(i,j) \in E}$ denote the messages at the $k$-th iteration of the algorithm. For an edge $(i,j)$ chosen uniformly at random, let $\mathbf{x}_q^{(k)}$ denote the random variable corresponding to the message $x_{i \to j}^{(k)}$ conditioned on the $i$-th task's difficulty being $q$. Similarly, let $\mathbf{y}_p^{(k)}$ denote the random variable corresponding to the message $y_{j \to i}^{(k)}$ conditioned on the $j$-th worker's quality being $p$.

At the first iteration, the task messages are updated according to $x_{i \to j}^{(1)} = \sum_{j' \in \partial i \setminus j} \mathbf{A}_{ij'} y_{j' \to i}^{(0)}$. Since we initialize the worker messages $\{y_{j \to i}^{(0)}\}_{(i,j) \in E}$ with independent Gaussian random variables with mean and variance both one, if we know the distribution of $\mathbf{A}_{ij'}$'s, then we have the distribution of $x_{i \to j}^{(1)}$. Since, we are assuming that the local subgraph is tree-like, all $x_{i \to j}^{(1)}$ for $i \in \mathbf{G}_{\mathbf{I},2k-1}$ for any randomly chosen node $\mathbf{I}$ are independent. Further, because of the symmetry in the construction of the random graph $\mathbf{G}$ all messages $x_{i \to j}^{(1)}$'s are identically distributed. Precisely, $x_{i \to j}^{(1)}$ are distributed according to $\mathbf{x}_q^{(1)}$ defined in Equation (28). In the following, we recursively define $\mathbf{x}_q^{(k)}$ and $\mathbf{y}_p^{(k)}$ in Equations (28) and (29).

For brevity, here and after, we drop the superscript $k$-iteration number whenever it is clear from the context. Let $\mathbf{x}_{q,a}$'s and $\mathbf{y}_{p,b}$'s be independent random variables distributed according to $\mathbf{x}_q$ and $\mathbf{y}_p$ respectively. We use $a$ and $b$ as indices for independent random variables with the same distribution. Also, let $\mathbf{z}_{p,q,a}$'s and $\mathbf{z}_{p,q,b}$'s be independent random variables distributed according to $\mathbf{z}_{p,q}$, where

$$\mathbf{z}_{p,q} = \begin{cases} +1 & \text{w.p.} \quad pq + (1-p)(1-q), \\ -1 & \text{w.p.} \quad p(1-q) + (1-p)q. \end{cases} \tag{27}$$

This represents the response given by a worker conditioned on the task having difficulty $q$ and the worker having ability $p$. Let $\mathscr{F}_1$ and $\mathscr{F}_2$ over $[0,1]$ be the distributions of the tasks' difficulty level and workers' quality respectively. Let $\mathbf{q} \sim \mathscr{F}_1$ and $\mathbf{p} \sim \mathscr{F}_2$. Then $\mathbf{q}_a$'s and $\mathbf{p}_b$'s are independent random variables distributed according to $\mathbf{q}$ and $\mathbf{p}$ respectively. Further, $\mathbf{z}_{p,\mathbf{q}_a,a}$'s and $\mathbf{x}_{\mathbf{q}_a,a}$'s are conditionally independent conditioned on $\mathbf{q}_a$; and $\mathbf{z}_{\mathbf{p}_b,q,b}$'s and $\mathbf{y}_{\mathbf{p}_b,b}$'s are conditionally independent conditioned on $\mathbf{p}_b$.

Let $\stackrel{d}{=}$ denote equality in distribution. Then for $k \in \{1, 2, \cdots\}$, the task messages (conditioned on the latent task difficulty level $q$) are distributed as the sum of $\ell - 1$ incoming messages that are i.i.d. according to $\mathbf{y}_{\mathbf{p}}^{(k-1)}$ and weighted by i.i.d. responses:

$$\mathbf{x}_q^{(k)} \stackrel{d}{=} \sum_{b \in [\ell-1]} \mathbf{z}_{\mathbf{p}_b, q, b} \mathbf{y}_{\mathbf{p}_b, b}^{(k-1)}. \tag{28}$$

Similarly, the worker messages (conditioned on the latent worker quality $p$) are distributed as the sum of $r - 1$ incoming messages that are i.i.d. according to $\mathbf{x}_{\mathbf{q}}^{(k)}$ and weighted by the i.i.d. responses:

$$\mathbf{y}_p^{(k)} \stackrel{d}{=} \sum_{a \in [r-1]} \mathbf{z}_{p, \mathbf{q}_a, a} \mathbf{x}_{\mathbf{q}_a, a}^{(k)}. \tag{29}$$

For the decision variable $\mathbf{x}_{\mathbf{I}}^{(k)}$ on a task $\mathbf{I}$ chosen uniformly at random, we have

$$\hat{\mathbf{x}}_q^{(k)} \stackrel{d}{=} \sum_{a \in [\ell]} \mathbf{z}_{\mathbf{p}_a, q, a} \mathbf{y}_{\mathbf{p}_a, a}^{(k-1)}. \tag{30}$$

**Mean and Variance Computation.** Define $m_{\mathbf{q}}^{(k)} \equiv \mathbb{E}[\mathbf{x}_{\mathbf{q}}^{(k)} | \mathbf{q}]$ and $\hat{m}_{\mathbf{p}}^{(k)} \equiv \mathbb{E}[\mathbf{y}_{\mathbf{p}}^{(k)} | \mathbf{p}]$, $\nu_{\mathbf{q}}^{(k)} \equiv \mathrm{Var}(\mathbf{x}_{\mathbf{q}}^{(k)} | \mathbf{q})$ and $\hat{\nu}_{\mathbf{p}}^{(k)} \equiv \mathrm{Var}(\mathbf{y}_{\mathbf{p}}^{(k)} | \mathbf{p})$. Recall the notations $\mu \equiv \mathbb{E}[2\mathbf{p} - 1]$, $\alpha \equiv \mathbb{E}[(2\mathbf{q} - 1)^2]$, $\beta \equiv \mathbb{E}[(2\mathbf{p} - 1)^2]$, $\hat{\ell} = \ell - 1$, and $\hat{r} = r - 1$. Then from (28) and (29) and using $\mathbb{E}[\mathbf{z}_{p,q}] = (2p - 1)(2q - 1)$ we get the following:

$$m_{\mathbf{q}}^{(k)} = \hat{\ell}(2\mathbf{q} - 1)\mathbb{E}_{\mathbf{p}}\left[(2\mathbf{p} - 1)\hat{m}_{\mathbf{p}}^{(k-1)}\right], \tag{31}$$

$$\hat{m}_{\mathbf{p}}^{(k)} = \hat{r}(2\mathbf{p} - 1)\mathbb{E}_{\mathbf{q}}\left[(2\mathbf{q} - 1)m_{\mathbf{q}}^{(k)}\right], \tag{32}$$

$$\nu_{\mathbf{q}}^{(k)} = \hat{\ell}\left\{\mathbb{E}_{\mathbf{p}}\left[\hat{\nu}_{\mathbf{p}}^{(k-1)} + (\hat{m}_{\mathbf{p}}^{(k-1)})^2\right] - (m_{\mathbf{q}}^{(k)}/\hat{\ell})^2\right\}, \tag{33}$$

$$\hat{\nu}_{\mathbf{p}}^{(k)} = \hat{r}\left\{\mathbb{E}_{\mathbf{q}}\left[\nu_{\mathbf{q}}^{(k)} + (m_{\mathbf{q}}^{(k)})^2\right] - (\hat{m}_{\mathbf{p}}^{(k)}/\hat{r})^2\right\}. \tag{34}$$

Define $m^{(k)} \equiv \mathbb{E}_{\mathbf{q}}[(2\mathbf{q} - 1)m_{\mathbf{q}}^{(k)}]$ and $\nu^{(k)} \equiv \mathbb{E}_{\mathbf{q}}[\nu_{\mathbf{q}}^{(k)}]$. From (31) and (32), we have the following recursion on the first moment of the random variable $x_q^{(k)}$:

$$m_{\mathbf{q}}^{(k)} = \hat{\ell}\hat{r}(2\mathbf{q} - 1)\beta m^{(k-1)}, m^{(k)} = \hat{\ell}\hat{r}\alpha\beta m^{(k-1)}. \tag{35}$$

From (33) and (34), and using $\mathbb{E}_{\mathbf{q}}[(m_{\mathbf{q}}^{(k)})^2] = (m^{(k)})^2/\alpha$ (from (35)), and $\mathbb{E}_{\mathbf{p}}[(\hat{m}_{\mathbf{p}}^{(k)})^2] = \hat{r}^2\beta(m^{(k)})^2$ (from (32)), we get the following recursion on the second moment:

$$\nu_{\mathbf{q}}^{(k)} = \hat{\ell}\hat{r}\nu^{(k-1)} + \hat{\ell}\hat{r}(m^{(k-1)})^2\left((1 - \alpha\beta)(1 + \hat{r}\alpha\beta) + \hat{r}\alpha(\beta)^2(\alpha - (2\mathbf{q} - 1)^2)\right)/\alpha, \tag{36}$$

$$\nu^{(k)} = \hat{\ell}\hat{r}\nu^{(k-1)} + \hat{\ell}\hat{r}(m^{(k-1)})^2(1 - \alpha\beta)(1 + \hat{r}\alpha\beta)/\alpha. \tag{37}$$

Since $\hat{m}_{\mathbf{p}}^{(0)} = 1$ as per our assumption, we have $m_{\mathbf{q}}^{(1)} = \hat{\ell}\mu(2\mathbf{q} - 1)$ and $m^{(1)} = \hat{\ell}\mu\alpha$. Therefore from (35), we have $m^{(k)} = \hat{\ell}\mu\alpha(\hat{\ell}\hat{r}\alpha\beta)^{k-1}$ and $m_{\mathbf{q}}^{(k)} = \hat{\ell}\mu(2\mathbf{q} - 1)(\hat{\ell}\hat{r}\alpha\beta)^{k-1}$. Further, since $\hat{\nu}_{\mathbf{p}}^{(0)} = 1$ as per our assumption, we have $\nu_{\mathbf{q}}^{(1)} = \hat{\ell}(2 - \mu^2(2\mathbf{q} - 1)^2)$ and $\nu^{(1)} = \hat{\ell}(2 - \mu^2\alpha)$. This implies that $\nu^{(k)} = a\nu^{(k-1)} + bc^{k-2}$, with $a = \hat{\ell}\hat{r}$, $b = \mu^2\alpha\hat{\ell}^3\hat{r}(1 - \alpha\beta)(1 + \hat{r}\alpha\beta)$ and $c = (\hat{\ell}\hat{r}\alpha\beta)^2$. After some algebra, we have that $\nu^{(k)} = \nu^{(1)}a^{k-1} + bc^{k-2}\sum_{\ell=0}^{k-2}(a/c)^{\ell}$. For $\hat{\ell}\hat{r}(\alpha\beta)^2 > 1$, we have $a/c < 1$ and

$$\nu_{\mathbf{q}}^{(k)} = \hat{\ell}(2 - \mu^2\alpha)(\hat{\ell}\hat{r})^{k-1} + \mu^2\hat{\ell}(\hat{\ell}\hat{r}\alpha\beta)^{2k-2}(\alpha - (2\mathbf{q} - 1)^2)$$
$$+ \left(\frac{1 - 1/(\hat{\ell}\hat{r}(\alpha\beta)^2)^{k-1}}{\hat{\ell}\hat{r}\alpha^2\beta^2 - 1}\right)(1 - \alpha\beta)(1 + \hat{r}\alpha\beta)\mu^2\alpha\hat{\ell}^2(\hat{\ell}\hat{r}\alpha\beta)^{2k-2}. \tag{38}$$

By a similar analysis, mean and variance of the decision variable $\hat{\mathbf{x}}_{\mathbf{q}}^{(k)}$ in (30) can also be computed. In particular, they are $\ell/\hat{\ell}$ times $m_{\mathbf{q}}^{(k)}$ and $\nu_{\mathbf{q}}^{(k)}$. Gaussianity of the messages follows due to Central limit theorem.

# 6  Proof of Theorem 2.3

The proof uses the results derived in the proof of Lemma 5.1.

Let $\hat{t}_i^{(k)}$ denote the resulting estimate of task $i$ after running the iterative inference algorithm for $k$ iterations. We want to compute the conditional probability of error of a task $\mathbf{I}$ selected uniformly at random in $[m]$, conditioned on its difficulty level, i.e.,

$$\mathbb{P}\big[t_{\mathbf{I}} \neq \hat{t}_{\mathbf{I}}^{(k)} \big| q_{\mathbf{I}}\big] \ .$$

In the following, we assume $q_{\mathbf{I}} \geq (1/2)$, i.e. the true label is $t_i = 1$. Analysis for $q_{\mathbf{I}} \leq (1/2)$ would be similar and result in the same bounds. Using the arguments given in Lemma 5.1, we have,

$$\mathbb{P}\big[t_{\mathbf{I}} \neq \hat{t}_{\mathbf{I}}^{(k)} \big| q_{\mathbf{I}}\big] \quad \leq \quad \mathbb{P}\big[t_{\mathbf{I}} \neq \hat{t}_{\mathbf{I}}^{(k)} \big| \mathbf{G}_{\mathbf{I},2k-1} \text{ is a tree, } q_{\mathbf{I}}\big] + \mathbb{P}\big[\mathbf{G}_{\mathbf{I},2k-1} \text{ is not a tree}\big]. \quad (39)$$

To provide an upper bound on the first term in (39), let $x_i^{(k)}$ denote the decision variable for task $i$ after $k$ iterations of the algorithm such that $\hat{t}_i^{(k)} = \text{sign}(x_i^{(k)})$. Then as per our assumption that $t_i = 1$, we have,

$$\mathbb{P}\big[t_{\mathbf{I}} \neq \hat{t}_{\mathbf{I}}^{(k)} \big| \mathbf{G}_{\mathbf{I},2k-1} \text{is a tree}, q_{\mathbf{I}}\big] \quad \leq \quad \mathbb{P}\big[x_{\mathbf{I}}^{(k)} \leq 0 \big| \mathbf{G}_{\mathbf{I},2k-1} \text{is a tree}, q_{\mathbf{I}}\big]. \quad (40)$$

Next, we apply "density evolution" [13] and provide a sharp upper bound on the probability of the decision variable $x_{\mathbf{I}}^{(k)}$ being negative in a locally tree like graph given $q_{\mathbf{I}} \geq (1/2)$. The proof technique is similar to the one introduced in [9]. Precisely, we show,

$$\mathbb{P}\big[x_{\mathbf{I}}^{(k)} \leq 0 \big| \mathbf{G}_{\mathbf{I},2k-1} \text{ is a tree }, q_{\mathbf{I}}\big] = \mathbb{P}\big[\hat{\mathbf{x}}_q^{(k)} \leq 0\big] \ , \quad (41)$$

where $\hat{\mathbf{x}}_q^{(k)}$ is defined in Equations (28)-(30) using density evolution. We will prove in the following that when $\hat{\ell}\hat{r}(\alpha\beta)^2 > 1$ and $\hat{r}\alpha > 1$,

$$\mathbb{P}\big[\hat{\mathbf{x}}_q^{(k)} \leq 0\big] \quad \leq \quad e^{-\ell\beta(2q_{\mathbf{I}}-1)^2/(2\sigma_k^2)}. \quad (42)$$

Theorem 2.3 follows by combining Equations (39),(26),(40) and (41).

we show that $\hat{\mathbf{x}}^{(k)}$ is sub-Gaussian with some appropriate parameter and then apply the Chernoff bound. A random variable $\mathbf{x}$ with mean $\mu$ is said to be *sub-Gaussian* with parameter $\sigma$ if for all $\lambda \in \mathbb{R}$ the following bound holds for its moment generating function:

$$\mathbb{E}[e^{\lambda\mathbf{x}}] \quad \leq \quad e^{\mu\lambda+(1/2)\sigma^2\lambda^2} \ . \quad (43)$$

Define,

$$\tilde{\sigma}_k^2 \equiv 3\hat{\ell}^3\hat{r}\mu^2\alpha(\hat{r}\alpha\beta+1)(\hat{\ell}\hat{r}\alpha\beta)^{2k-4}\Big(\frac{1-1/(\hat{\ell}\hat{r}(\alpha\beta)^2)^{k-1}}{1-1/(\hat{\ell}\hat{r}\alpha\beta)}\Big) + 2\hat{\ell}(\hat{\ell}\hat{r})^{k-1} \ , \quad (44)$$

$m_k \equiv \mu\hat{\ell}(\hat{\ell}\hat{r}\alpha\beta)^{k-1}$, and $m_{k,\mathbf{q}} \equiv (2\mathbf{q}-1)m_k$ for $k \in \mathbb{Z}$, where $\mathbf{q} \sim \mathscr{F}_1$. We will show that, $\mathbf{x}_{\mathbf{q}}^{(k)}$ is sub-Gaussian with mean $m_{k,\mathbf{q}}$ and parameter $\tilde{\sigma}_k^2$ for $|\lambda| \leq 1/(2m_{k-1}\hat{r}\alpha)$, i.e.,

$$\mathbb{E}[e^{\lambda\mathbf{x}_{\mathbf{q}}^{(k)}}|\mathbf{q}] \quad \leq \quad e^{m_{k,\mathbf{q}}\lambda+(1/2)\tilde{\sigma}_k^2\lambda^2} \ . \quad (45)$$

**Analyzing the Density.** Notice that the parameter $\tilde{\sigma}_k^2$ does not depend upon the random variable $\mathbf{q}$. By definition of $\hat{\mathbf{x}}_{\mathbf{q}}^{(k)}$, (30), we have $\mathbb{E}[e^{\lambda\hat{\mathbf{x}}_{\mathbf{q}}^{(k)}}|\mathbf{q}] = \mathbb{E}[e^{\lambda\mathbf{x}_{\mathbf{q}}^{(k)}}|\mathbf{q}]^{(\ell/\hat{\ell})}$. Therefore, it follows that $\mathbb{E}[e^{\lambda\hat{\mathbf{x}}_{\mathbf{q}}^{(k)}}|\mathbf{q}] \leq e^{(\ell/\hat{\ell})m_{k,\mathbf{q}}\lambda+(\ell/2\hat{\ell})\tilde{\sigma}_k^2\lambda^2}$. Using the Chernoff bound with $\lambda = -m_{k,\mathbf{q}}/(\tilde{\sigma}_k^2)$, we have

$$\mathbb{P}[\hat{\mathbf{x}}_{\mathbf{q}}^{(k)} \leq 0 \mid \mathbf{q}] \quad \leq \quad \mathbb{E}[e^{\lambda\hat{\mathbf{x}}_{\mathbf{q}}^{(k)}}|\mathbf{q}] \quad \leq \quad e^{-\ell m_{k,\mathbf{q}}^2/(2\hat{\ell}\tilde{\sigma}_k^2)} \ . \quad (46)$$

Note that, with the assumption that $\mathbf{q} \geq (1/2)$, $m_{k,\mathbf{q}}$ is non-negative. Since

$$\frac{m_{k,\mathbf{q}}m_{k-1,\mathbf{q}}}{\tilde{\sigma}_k^2} \leq \frac{(2\mathbf{q}-1)^2\mu^2\hat{\ell}^2(\hat{\ell}\hat{r}\alpha\beta)^{2k-3}}{3\mu^2\beta(\alpha)^2\hat{\ell}^3\hat{r}^2(\hat{\ell}\hat{r}\alpha\beta)^{2k-4}} = \frac{(2\mathbf{q}-1)^2}{3\hat{r}\alpha} \ ,$$

it follows that $|\lambda| \leq 1/(2m_{k-1}\hat{r}\alpha)$. The desired bound in (42) follows.

Now, we are left to prove Equation (45). From (28) and (29), we have the following recursive formula for the evolution of the moment generating functions of $\mathbf{x_q}$ and $\mathbf{y_p}$:

$$\mathbb{E}[e^{\lambda \mathbf{x_q}^{(k)}}|\mathbf{q}] = \left(\mathbb{E}_{\mathbf{p}}\left[(\mathbf{pq} + \bar{\mathbf{p}}\bar{\mathbf{q}})\mathbb{E}[e^{\lambda \mathbf{y_p}^{(k-1)}}|\mathbf{p}] + (\mathbf{p}\bar{\mathbf{q}} + \bar{\mathbf{p}}\mathbf{q})\mathbb{E}[e^{-\lambda \mathbf{y_p}^{(k-1)}}|\mathbf{p}]\right]\right)^{\hat{\ell}}, \quad (47)$$

$$\mathbb{E}[e^{\lambda \mathbf{y_p}^{(k)}}|\mathbf{p}] = \left(\mathbb{E}_{\mathbf{q}}\left[(\mathbf{pq} + \bar{\mathbf{p}}\bar{\mathbf{q}})\mathbb{E}[e^{\lambda \mathbf{x_q}^{(k)}}|\mathbf{q}] + (\mathbf{p}\bar{\mathbf{q}} + \bar{\mathbf{p}}\mathbf{q})\mathbb{E}[e^{-\lambda \mathbf{x_q}^{(k)}}|\mathbf{q}]\right]\right)^{\hat{r}}, \quad (48)$$

where $\bar{\mathbf{p}} = 1 - \mathbf{p}$ and $\bar{\mathbf{q}} = 1 - \mathbf{q}$. We apply induction to prove that the messages are sub-Gaussian. First, for $k = 1$, we show that $\mathbf{x_q}^{(1)}$ is sub-Gaussian with mean $m_{1,\mathbf{q}} = (2\mathbf{q} - 1)\mu\hat{\ell}$ and parameter $\tilde{\sigma}_1^2 = 2\hat{\ell}$. Since, $\mathbf{y_p}$ is initialized as a Gaussian random variable with mean and variance both one, we have $\mathbb{E}[e^{\lambda \mathbf{y_p}^{(0)}}] = e^{\lambda + (1/2)\lambda^2}$. Substituting this into Equation (47), we get for any $\lambda$,

$$\mathbb{E}[e^{\lambda \mathbf{x_q}^{(1)}}|\mathbf{q}] = \left((\mathbb{E}[\mathbf{p}]\mathbf{q} + \mathbb{E}[\bar{\mathbf{p}}]\bar{\mathbf{q}})e^{\lambda} + (\mathbb{E}[\mathbf{p}]\bar{\mathbf{q}} + \mathbb{E}[\bar{\mathbf{p}}]\mathbf{q})e^{-\lambda}\right)^{\hat{\ell}} e^{(1/2)\lambda^2 \hat{\ell}} \quad (49)$$

$$\leq e^{(2\mathbf{q}-1)\mu\hat{\ell}\lambda + (1/2)(2\hat{\ell})\lambda^2}, \quad (50)$$

where the inequality follows from the fact that $ae^z + (1-a)e^{-z} \leq e^{(2a-1)z + (1/2)z^2}$ for any $z \in \mathbb{R}$ and $a \in [0, 1]$ (Lemma A.1.5 from [1]). Next, assuming $\mathbb{E}[e^{\lambda \mathbf{x_q}^{(k)}}|\mathbf{q}] \leq e^{m_{k,\mathbf{q}}\lambda + (1/2)\tilde{\sigma}_k^2\lambda^2}$ for $|\lambda| \leq 1/(2m_{k-1}\hat{r}\alpha)$, we show that $\mathbb{E}[e^{\lambda \mathbf{x_q}^{(k+1)}}|\mathbf{q}] \leq e^{m_{k+1,\mathbf{q}}\lambda + (1/2)\tilde{\sigma}_{k+1}^2\lambda^2}$ for $|\lambda| \leq 1/(2m_k\hat{r}\alpha)$, and compute appropriate $m_{k+1,\mathbf{q}}$ and $\tilde{\sigma}_{k+1}^2$.

Substituting the bound $\mathbb{E}[e^{\lambda \mathbf{x_q}^{(k)}}|\mathbf{q}] \leq e^{m_{k,\mathbf{q}}\lambda + (1/2)\tilde{\sigma}_k^2\lambda^2}$ in (48), we have

$$\mathbb{E}[e^{\lambda \mathbf{y_p}^{(k)}}|\mathbf{p}]$$

$$\leq \left(\mathbb{E}_{\mathbf{q}}\left[(\mathbf{pq} + \bar{\mathbf{p}}\bar{\mathbf{q}})e^{m_{k,\mathbf{q}}\lambda} + (\mathbf{p}\bar{\mathbf{q}} + \bar{\mathbf{p}}\mathbf{q})e^{-m_{k,\mathbf{q}}\lambda}\right]\right)^{\hat{r}} e^{(1/2)\tilde{\sigma}_k^2\lambda^2\hat{r}}$$

$$\leq \left(\mathbb{E}_{\mathbf{q}}\left[e^{(2\mathbf{q}-1)(2\mathbf{p}-1)m_{k,\mathbf{q}}\lambda + (1/2)(m_{k,\mathbf{q}}\lambda)^2}\right]\right)^{\hat{r}} e^{(1/2)\tilde{\sigma}_k^2\lambda^2\hat{r}} \quad (51)$$

$$= \left(\mathbb{E}_{\mathbf{q}}\left[e^{(2\mathbf{p}-1)(2\mathbf{q}-1)^2 m_k\lambda + (1/2)(2\mathbf{q}-1)^2 (m_k\lambda)^2}\right]\right)^{\hat{r}} e^{0.5\tilde{\sigma}_k^2\lambda^2\hat{r}} \quad (52)$$

where (51) uses the inequality $ae^z + (1-a)e^{-z} \leq e^{(2a-1)z + (1/2)z^2}$ and (52) follows from the definition of $m_{k,\mathbf{q}} \equiv (2\mathbf{q} - 1)m_k$. To bound the term in (52), we use the following lemma.

**Lemma 6.1.** *For any random variable $s \in [0, 1]$, $|z| \leq 1/2$ and $|t| < 1$, we have*

$$\mathbb{E}\left[e^{stz + (1/2)sz^2}\right] \leq \exp\left(\mathbb{E}[s]tz + (3/2)\mathbb{E}[s]z^2\right). \quad (53)$$

For $|\lambda| \leq 1/(2m_k\hat{r}\alpha)$, using the assumption that $\hat{r}\alpha > 1$, we have $m_k\lambda \leq (1/2)$. Applying Lemma 6.1 on the term in (52), with $s = (2\mathbf{q} - 1)^2$, $z = m_k\lambda$ and $t = (2p - 1)$, we get

$$\mathbb{E}[e^{\lambda \mathbf{y_p}^{(k)}}|\mathbf{p}] \leq e^{\alpha(2\mathbf{p}-1)\hat{r}m_k\lambda + (1/2)\left(3\alpha m_k^2 + \tilde{\sigma}_k^2\right)\lambda^2\hat{r}}. \quad (54)$$

Substituting the bound in (54) in Equation (47), we get

$$\mathbb{E}[e^{\lambda \mathbf{x_q}^{(k+1)}}|\mathbf{q}]$$

$$\leq \left(\mathbb{E}_{\mathbf{p}}\left[(\mathbf{pq} + \bar{\mathbf{p}}\bar{\mathbf{q}})e^{\alpha(2\mathbf{p}-1)m_k\lambda\hat{r}} + (\mathbf{p}\bar{\mathbf{q}} + \bar{\mathbf{p}}\mathbf{q})e^{-\alpha(2\mathbf{p}-1)m_k\lambda\hat{r}}\right]\right)^{\hat{\ell}} e^{(1/2)(3\alpha m_k^2 + \tilde{\sigma}_k^2)\lambda^2\hat{\ell}\hat{r}}$$

$$\leq \left(\mathbb{E}_{\mathbf{p}}\left[e^{(2\mathbf{q}-1)(2\mathbf{p}-1)^2\alpha m_k\lambda\hat{r} + (1/2)(2\mathbf{p}-1)^2(\alpha m_k\lambda\hat{r})^2}\right]\right)^{\hat{\ell}} e^{(1/2)(3\alpha m_k^2 + \tilde{\sigma}_k^2)\lambda^2\hat{\ell}\hat{r}} \quad (55)$$

$$\leq e^{\hat{\ell}\hat{r}\alpha\beta m_{k,\mathbf{q}}\lambda + (1/2)\hat{\ell}\hat{r}\left(\tilde{\sigma}_k^2 + 3\alpha m_k^2(1 + \hat{r}\alpha\beta)\right)\lambda^2}, \quad (56)$$

where (55) uses the inequality $ae^z + (1-a)e^{-z} \leq e^{(2a-1)z + (1/2)z^2}$. Equation (56) follows from the application of Lemma 6.1, with $s = (2\mathbf{p}-1)^2$, $z = \alpha m_k\lambda\hat{r}$ and $t = (2\mathbf{q}-1)$. For $|\lambda| \leq 1/(2m_k\hat{r}\alpha)$, $|z| < (1/2)$.

In the regime where $\hat{\ell}\hat{r}(\alpha\beta)^2 > 1$, as per our assumption, $m_k$ is non-decreasing in $k$. At iteration $k$, the above recursion holds for $|\lambda| \leq 1/(2\hat{r}\alpha)\min\{1/m_1, \cdots, 1/m_{k-1}\} = 1/(2m_{k-1}\hat{r}\alpha)$. Hence, we get the following recursion for $m_{k,\mathbf{q}}$ and $\tilde{\sigma}_k^2$ such that (45) holds for $|\lambda| \leq 1/(2m_{k-1}\hat{r}\alpha)$:

$$m_{k,\mathbf{q}} = \hat{\ell}\hat{r}\alpha\beta m_{k-1,\mathbf{q}},$$
$$\tilde{\sigma}_k^2 = \hat{\ell}\hat{r}\tilde{\sigma}_{k-1}^2 + 3\hat{\ell}\hat{r}(1 + \hat{r}\alpha\beta)\alpha m_{k-1}^2. \quad (57)$$

With the initialization $m_{1,\mathbf{q}} = (2\mathbf{q}-1)\mu\hat{\ell}$ and $\tilde{\sigma}_1^2 = 2\hat{\ell}$, we have $m_{k,\mathbf{q}} = \mu(2\mathbf{q}-1)\hat{\ell}(\alpha\beta\hat{\ell}\hat{r})^{k-1}$ for $k \in \{1,2,\cdots\}$ and $\tilde{\sigma}_k^2 = a\tilde{\sigma}_{k-1}^2 + bc^{k-2}$ for $k \in \{2,3\cdots\}$, with $a = \hat{\ell}\hat{r}$, $b = 3\hat{\ell}^3\hat{r}\mu^2\alpha(1+\alpha\beta\hat{r})$, and $c = (\alpha\beta\hat{\ell}\hat{r})^2$. After some algebra, we have $\tilde{\sigma}_k^2 = \tilde{\sigma}_1^2 a^{k-1} + bc^{k-2}\sum_{\ell=0}^{k-2}(a/c)^\ell$. For $\hat{\ell}\hat{r}(\alpha\beta)^2 \neq 1$, we have $a/c \neq 1$, whence $\tilde{\sigma}_k^2 = \tilde{\sigma}_1^2 a^{k-1} + bc^{k-2}(1-(a/c)^{k-1})/(1-a/c)$. This finishes the proof of (45).

## 6.1 Proof of Lemma 6.1

Using the fact that $e^a \leq 1 + a + 0.63a^2$ for $|a| \leq 5/8$,

$$
\begin{aligned}
&\mathbb{E}\left[e^{stz+(1/2)sz^2}\right] \\
\leq\ &\mathbb{E}\left[1 + stz + (1/2)sz^2 + 0.63\left(stz + (1/2)sz^2\right)^2\right] \\
\leq\ &\mathbb{E}\left[1 + stz + (1/2)sz^2 + 0.63\left((5/4)z\sqrt{s}\right)^2\right] \\
\leq\ &1 + \mathbb{E}[s]tz + (3/2)\mathbb{E}[s]z^2 \\
\leq\ &\exp\left(\mathbb{E}[s]tz + (3/2)\mathbb{E}[s]z^2\right).
\end{aligned}
$$

# 7 Proof of Theorem 2.4

Let $\mathscr{F}$ denote a distribution on the worker quality $\mathbf{p}_j$ such that $\mathbf{p}_j \sim \mathscr{F}$. Let $\mathscr{F}_\beta$ be a collection of all distributions $\mathscr{F}$ such that:

$$
\mathscr{F}_\beta = \left\{\mathscr{F} \mid \mathbb{E}_{\mathscr{F}}[(2\mathbf{p}_j - 1)^2] = \beta\right\}.
$$

Define the minimax rate on the probability of error of a task $i$, conditioned on its difficulty level $q_i$, as

$$
\min_{\tau \in \mathscr{T}_{\ell_i}, \hat{t}} \ \max_{t_i \in \{\pm\}, \mathscr{F} \in \mathscr{F}_\beta} \mathbb{P}[t_i \neq \hat{t}_i \mid q_i], \tag{58}
$$

where $\mathscr{T}_{\ell_i}$ is the set of all nonadaptive task assignment schemes that assign $\ell_i$ workers to task $i$, and $\hat{t}$ ranges over the set of all estimators of $t_i$. Since the minimax rate is the maximum over all the distributions $\mathscr{F} \in \mathscr{F}_\beta$, we consider a particular worker quality distribution to get a lower bound on it. In particular, we assume the $\mathbf{p}_j$'s are drawn from a spammer-hammer model with perfect hammers:

$$
\mathbf{p}_j = \begin{cases} 1/2 & \text{with probability } 1-\beta, \\ 1 & \text{otherwise.} \end{cases}
$$

Observe that the chosen spammer-hammer models belongs to $\mathscr{F}_\beta$, i.e. $\mathbb{E}[(2\mathbf{p}_j - 1)^2] = \beta$. To get the optimal estimator, we consider an oracle estimator that knows all the $\mathbf{p}_j$'s and hence makes an optimal estimation. It estimates $\hat{t}_i$ using majority voting on hammers and ignores the answers of hammers. If there are no hammers then it flips a fair coin and estimates $\hat{t}_i$ correctly with half probability. It does the same in case of tie among the hammers. Concretely,

$$
\hat{t}_i = \text{sign}\left(\sum_{j \in W_i} \mathbb{I}\{j \in \mathbb{H}\})A_{ij}\right),
$$

where $W_i$ denotes the neighborhood of node $i$ in the graph and $\mathbb{H}$ is the set of hammers. Note that this is the optimal estimation for the spammer-hammer model. We want to compute a lower bound on $\mathbb{P}[t_i \neq \hat{t}_i | q_i]$. Let $\tilde{\ell}_i$ be the number of hammers answering task $i$, i.e.,$\tilde{\ell}_i = |W_i \cap \mathbb{H}|$. Since $p_j$'s are drawn from spammer-hammer model, $\tilde{\ell}_i$ is a binomial random variable $\text{Binom}(\ell_i, \beta)$. We first compute probability of error conditioned on $\tilde{\ell}_i$, i.e. $\mathbb{P}[t_i \neq \hat{t}_i | \tilde{\ell}_i, q_i]$. For this, we use the following lemma from [9].

**Lemma 7.1** (Lemma 2 from [9]). *For any $C < 1$, there exists a positive constant $C'$ such that when $(2q_i - 1) \leq C$, the error achieved by majority voting is at least*

$$
\min_{\tau \in \mathscr{T}_{\tilde{\ell}}} \ \max_{t_i \in \{\pm\}} \mathbb{P}[t_i \neq \hat{t}_i | \tilde{\ell}_i, q_i] \geq e^{-C'(\tilde{\ell}_i(2q_i-1)^2+1)}. \tag{59}
$$

Taking expectation with respect to random variable $\tilde{\ell}_i$ and applying Jensen's inequality on the term in right side, we get a lower bound on the minimiax probability of error in (58)

$$\min_{\tau \in \mathscr{T}_{\tilde{\ell}}, \hat{t}} \max_{\substack{\mathscr{F} \in \mathscr{F}_\beta \\ t_i \in \{\pm\}}} \mathbb{P}[t_i \neq \hat{t}_i | q_i] \geq e^{-C'(\ell_i \beta(2q_i - 1)^2 + 1)} . \tag{60}$$