[Reviews · NeurIPS 2016]

Reviewer 1

Summary

The paper concerns schemes for allocating binary classification tasks to a pool of workers, to achieve a target error rate while minimizing the cost of obtaining labels from the workers. The paper presents an adaptive task allocation scheme whose budget requirement, to achieve a given level of classification error, is asymptotically optimal up to constant factors, and it shows that this improves the performance of the best non-adaptive scheme by a factor of \lambda/\lambda_{min}, where \lambda and \lambda_{min} are two instance-specific parameters whose ratio is, in some cases, super-constant.

Qualitative Assessment

The paper addresses a well-motivated problem at the interface of crowdsourcing and ML, has significant technical depth, and I feel I learned a lot from it. I believe it would make a valuable addition to the NIPS program. My only comments for the authors concern two spots where the exposition could possibly be improved. 1. On page 2, just before the "Related Work" paragraph, you write, "We study the minimax rate when the nature chooses the worst case priors F and G...". But this is not actually what you study. (And it would be misguided to study the worst case F and G, as that would unnecessarily focus attention on nearly impossible classification problems.) Instead you consider the worst case priors F and G subject to a specified average worker quality, \beta, and a specified average task difficulty, \lambda. 2. In Section 2.3 you present a lower bound on the budget required by non-adaptive task allocation schemes. However, if I am interpreting the paragraph after Lemma 2.3 correctly, you require that for all tasks (no matter how difficult) the scheme has probability at most \epsilon of mislabeling. To achieve average error rate \epsilon, however, I suspect it would be budget-optimal to give up entirely on the goal of classifying a small fraction (say, \epsilon/2) of the most difficult tasks, and at least intuitively this logic leads me to suspect that the \lambda_{min} in the denominator on the right side of (10) should be replaced by the \epsilon-quantile of the distribution of \lambda_i. Or maybe the (\epsilon/2)-quantile. But I can't see why the reasoning given in Section 2.3 justifies a lower bound involving \lambda_{min}, when the objective of the non-adaptive task assignment scheme is merely to attain *average* error rate \epsilon, rather than probability of error \epsilon on each task individually. Am I misunderstanding something?

Confidence in this Review

2-Confident (read it all; understood it all reasonably well)


Reviewer 2

Summary

In crowdsourcing applications, the budget is limited and tasks have different difficulties. This work proposes an adaptive scheme for this problem. The proposed algorithm consists of multiple rounds. At each round, the algorithm uses a fraction of budget and decides answers for each task only if the quality of the collected information is above a threshold. They show the expected error of the adaptive algorithm, which is close to the lower bound what they find in Theorem 2.1.

Qualitative Assessment

This paper is very well written. Their idea is clear and their results look nice. I have only concern on Theorem 2.1. If I am wrong and the authors can make answer for my questions on the proof of Theorem 2.1, I'm very happy with this paper. "I have only concern on Theorem 2.1. I think the proof of Theorem 2.1 has a severe error. In adaptive algorithms, the number of workers W_i and the observed values have dependency. For instance, the proposed algorithm (Algorithm 1) hires more workers when the collected answers are not clear to decide. The authors cannot say Eq. (52) . Eq. (52) is true only if the algorithm hires W_i workers and then, collected information. However, adaptive algorithms hire more workers according to the collected information." ============= Thank you for the response! Now, I believe all theorems are correct!

Confidence in this Review

2-Confident (read it all; understood it all reasonably well)


Reviewer 3

Summary

This paper provides bounds on the gain of adaptive task allocation in crowdsourcing against non-adaptive allocation schemes. The authors provide a series of intuitive and very interesting bounds on the budget required to achieve a certain level of average accuracy over crowdsourcing schemes. Finally, the authors propose an adaptive task assignment scheme that matches the bounds introduced.

Qualitative Assessment

This is a very interesting paper to read. I found the proposed bounds and the theoretical analysis of the paper, elegant, easy to follow and quite insightful. The fact that the paper proposes an adaptive task assignment scheme that matches the proposed bounds is definitely a plus. A couple of comments thought. I believe the discussion around the algorithm on page 5 should be more modular so that you the reader can really understand it in detail. The experimental evaluation backs up the theoretical findings. Nonetheless, I wish the authors have evaluated their theoretical results on real-world data by simulating the adaptivity of the proposed scheme. Also the paper needs some polishing in terms of wording and structure. It feels weird that there is no conclusion section.

Confidence in this Review

2-Confident (read it all; understood it all reasonably well)


Reviewer 4

Summary

The paper looks at adaptive schemes in crowdsourcing and analyzes theoretically the trade-off between cost and accuracy under a generalized Dawid-Skene model. It also proposes an adaptive algorithm that achieves the theoretical limit and is strictly better than any non-adaptive algorithm.

Qualitative Assessment

The paper provides a theoretical analysis of adaptive schemes in crowdsourcing. This is quite interesting work as many papers about crowdsourcing do not give any theoretical guarantees for their proposed solutions. I am not very well-versed in the theoretical aspects mentioned in this paper, so I found it quite dense and had difficulty following the proofs. I will therefore focus in my comments on the aspects that would be of greater interest to a practitioner of crowdsourcing. The algorithm: The description/pseudocode is not easy to understand. The explanation in the text helps considerably, however some details are not explained (for example, why are X and s_t set the way they are?). Also, what is the running time? Is it impractical to run on large data sets, or is it fast enough? The simulation results focus on quite large numbers of queries per task. In practice it is uncommon to use more than 5 or 10 workers per task. A smaller range is not shown in the graphs (it's hard to say at what number they actually start, but looks like 20) and this range is also where the adaptive algorithm seems to perform the same or worse than the non-adaptive. Does this mean that the proposed algorithm is mainly of theoretical interest? Is it better in practice to stick with the non-adaptive version for small number of queries per task? It would have been nice to see the x-axis in the charts starting from 2 or 3 and go higher. Summary/conclusions section is missing. It should summarize what was achieved and give directions to other research that naturally follows these results. There are some problems in the references: - The formatting is mixed. There are authors with full first name and others with initial. There are titles missing (e.g. [15]). - Many references are not mentioned in the text: [1, 2, 5, 6, 9, 12, 13, 17, 20, 21] - [10] and [11] are the same reference. [15] and [16] are also probably the same reference. Other issues: - line 53: support - line 101: simulations confirm - both figures: the caption should explain what is on the left vs right - line 237: Theorem 2.4 is the incorrect number - line 250: at least - line 284: error rate ... is dominated - the supplementary material is out of order, theorem 2.2 is first, but 2.1 is last. - why does the proof of lemma 3.1 have its own section 4?

Confidence in this Review

1-Less confident (might not have understood significant parts)


Reviewer 5

Summary

Due to the potentially unreliable nature of crowdsourced workers(spammers or hammers), it is usually necessary to introduce redundancy in the tasks; i.e. offer the same task multiple times to different people. However, based on the difficulty of the task and the reliability of the workers, schemes which iteratively tune the redundancy are desirable. This is exactly the question being considered in the paper under the Dawid-Skene model, a recent probabilistic model to model the responses of the data/tasks. They actually work with a slight generalization of this model which allows for heterogenity in the tasks as well as the workers. First, a lower bound on the minimax error rate, the best error rate achievable by any adaptive scheme and any inference algorithm. This naturally leads to the redundancy parameter/ budget required to achieve an error rate of \epsilon. The authors then present an adaptive scheme and an inference algorithm (based on message passing) which asymptotically achieves this error rate. They also provide an achievable error rate in the non-adaptive case and a lower bound in that case. Experiments on synthetic data based on the spammer-hammer model are provided and the comparison is against non-adaptive schemes and majority voting. The experiments show that the algorithm presented by the authors does much better than the other two methods.

Qualitative Assessment

I liked reading the paper. Most of the required background was presented in a manner which was easy to follow and the ideas leading to their contributions also flowed naturally. I think this problem and the solution presented in the paper are certainly interesting for certain parts of the community. The proofs are also elegant and I think this paper is certainly of NIPS level. I just had one thing which I was not sure about : During the start of the algorithm, a fraction of the tasks are classified based on a certain criteria (line 169). I didn't find the criteria mentioned in the paper. I presume it doesn't matter due to the inference algorithm and given enough rounds but I was still curious about it and I think it should be clarified in the paper, if not already done. Also, it would be nice if there were also experiments on real-world crowdsourcing platforms so that we get a better idea about how useful this method is. Some minor comments: Line 53:Spelling of support Line 217, In Figure 1, we instead of We

Confidence in this Review

2-Confident (read it all; understood it all reasonably well)